# Integrated methylome and phenome study of the circulating proteome reveals markers pertinent to brain health

Danni A. Gadd [1], Robert F. Hillary [1], Daniel L. McCartney [1], Liu Shi[2], Aleks Stolicyn [3], Neil A. Robertson [4], Rosie M. Walker [5], Robert I. McGeachan [6,7], Archie Campbell [1], Shen Xueyi [3], Miruna C. Barbu[3], Claire Green [3], Stewart W. Morris[1], Mathew A. Harris [3], Ellen V. Backhouse[5], Joanna M. Wardlaw [5,8,9], J. Douglas Steele[10], Diego A. Oyarzún [11,12,13], Graciela Muniz-Terrera[14,15], Craig Ritchie[14], Alejo Nevado-Holgado[2], Tamir Chandra[4], Caroline Hayward [1,4], Kathryn L. Evans[1], David J. Porteous [1], Simon R. Cox [16,17], Heather C. Whalley [3], Andrew M. McIntosh [3] & Riccardo E. Marioni [1] ✉

Characterising associations between the methylome, proteome and phenome may provide insight into biological pathways governing brain health. Here, we report an integrated DNA methylation and phenotypic study of the circulating proteome in relation to brain health. Methylome-wide association studies of 4058 plasma proteins are performed ($N = 774$), identifying 2928 CpG-protein associations after adjustment for multiple testing. These are independent of known genetic protein quantitative trait loci (pQTLs) and common lifestyle effects. Phenome-wide association studies of each protein are then performed in relation to 15 neurological traits ($N = 1,065$), identifying 405 associations between the levels of 191 proteins and cognitive scores, brain imaging measures or *APOE* e4 status. We uncover 35 previously unreported DNA methylation signatures for 17 protein markers of brain health. The epigenetic and proteomic markers we identify are pertinent to understanding and stratifying brain health.

[1]Centre for Genomic and Experimental Medicine, Institute of Genetics and Cancer, University of Edinburgh, Edinburgh EH4 2XU, UK. [2]Department of Psychiatry, University of Oxford, Oxford OX3 7JX, UK. [3]Division of Psychiatry, University of Edinburgh, Royal Edinburgh Hospital, Edinburgh EH10 5HF, UK. [4]MRC Human Genetics Unit, Institute of Genetics and Cancer, University of Edinburgh, Edinburgh EH4 2XU, UK. [5]Centre for Clinical Brain Sciences, Chancellor's Building, 49 Little France Crescent, Edinburgh BioQuarter, Edinburgh EH16 4SB, UK. [6]Centre for Discovery Brain Sciences, University of Edinburgh, 1 George Square, Edinburgh EH8 9JZ, UK. [7]The Hospital for Small Animals, Royal (Dick) School of Veterinary Studies, The University of Edinburgh, Easter Bush Campus, Edinburgh EH25 9RG, UK. [8]Edinburgh Imaging, University of Edinburgh, Edinburgh, UK. [9]UK Dementia Research Institute, University of Edinburgh, Edinburgh EH8 9JZ, UK. [10]Division of Imaging Science and Technology, Medical School, University of Dundee, Dundee DD1 9SY, UK. [11]School of Informatics, University of Edinburgh, Edinburgh EH8 9AB, UK. [12]School of Biological Sciences, University of Edinburgh, Edinburgh EH3 3JF, UK. [13]The Alan Turing Institute, 96 Euston Road, London NW1 2DB, UK. [14]Centre for Clinical Brain Sciences, Edinburgh Dementia Prevention, University of Edinburgh, Edinburgh EH4 2XU, UK. [15]Department of Social Medicine, Ohio University, Athens, OH 45701, USA. [16]Lothian Birth Cohorts, University of Edinburgh, Edinburgh EH8 9JZ, UK. [17]Department of Psychology, University of Edinburgh, Edinburgh EH8 9JZ, UK. ✉e-mail: riccardo.marioni@ed.ac.uk

The health of the ageing brain is associated with risk of neurodegenerative disease[1,2]. Relative brain age—a measure of brain health calculated using multiple volumetric brain imaging measures—has recently been shown to predict the development of dementia[3]. Structural brain imaging and performance in cognitive tests are well-characterised markers of brain health[4], which clearly associate with potentially modifiable traits such as body mass index (BMI), smoking and diabetes[5–7]. Understanding the interplay between environment, biology and brain health may therefore inform preventative strategies.

Multiple layers of omics data indicate the biological pathways that underlie phenotypes. Proteomic blood sampling can track peripheral pathways that may impact brain health, or record proteins secreted from the brain into the circulatory system. Although proteome-wide characterisation of cognitive decline and dementia risk[8–10] have been facilitated at large scale by SOMAscan® protein measurements, there is a need to further integrate omics to characterise brain health phenotypes. Epigenetic modifications to the genome record an individual's response to environmental exposures, stochastic biological effects, and genetic influences. Epigenetic changes include histone modifications, non-coding RNA, chromatin remodelling, and DNA methylation (DNAm) at cytosine bases, such as 5-hydroxymethylcytosine. These are implicated in changes to chromatin structure and the regulation of pathways associated with neurological diseases[11,12]. However, DNAm at cytosine-guanine (CpG) dinucleotides is the most widely profiled blood-based epigenetic modification at large scale.

Modifications to DNAm at CpG sites play differential roles in influencing gene expression at the transcriptional level[13]. Additionally, DNAm accounts for inter-individual variability in circulating protein levels[14–16]. Recently, through integration of DNAm and protein data, we have shown that epigenetic scores for plasma protein levels—known as EpiScores—associate with brain morphology and cognitive ageing markers[17] and predict the onset of neurological diseases[18]. These studies highlight that while datasets that allow for integration of proteomic, epigenetic and phenotypic information are rarely-available, they hold potential to advance risk stratification. Integration may also uncover candidate biological pathways that may underlie brain health.

Associations between protein levels and DNAm at CpGs are known as protein quantitative trait methylation loci (pQTMs) and can be quantified by methylome-wide association studies (MWAS) of protein levels. The largest MWAS of protein levels to date assessed 1123 SOMAmer protein measurements in the German KORA cohort ($n = 944$)[14]. In that study, Zaghlool et al. reported 98 pQTMs that replicated in the QMDiab cohort ($n = 344$), with significant associations

between DNAm at the immune-associated locus *NLRC5* and seven immune-related proteins ($P < 2.5 \times 10^{-7}$). This suggested that DNAm not only reflects variability in the proteome but is closely related to chronic systemic inflammation. Hillary et al. have also assessed epigenetic signatures for 281 SOMAmer protein measurements that were previously associated with Alzheimer's disease, in the Generation Scotland cohort that we utilised in this study[19]. However, proteome-wide assessment of pQTMs has not been tested against a comprehensive spectrum of brain health traits.

Here, we conduct an integrated methylome- and phenome-wide assessment of the circulating proteome in relation to brain health (Fig. 1), using 4058 protein level measurements (Annotation information provided in Supplementary Data 1). We characterise CpG–protein associations (pQTMs) for these proteins in 774 individuals from the Generation Scotland cohort using EPIC array DNAm at 772,619 CpG sites. We then identify which of the 4058 protein levels associate with one or more of 15 neurological traits (seven structural brain imaging measures, seven cognitive scores and *APOE* e4 status) in 1065 individuals from the same cohort where the pQTM data are a nested subset. By overlapping these datasets, we probe the epigenetic signatures of proteins that are related to brain health. For these signatures, we map potential underlying genetic components and chromatin interactions that may play a role in protein level regulation.

## Results
### Methylome-wide studies of 4058 plasma proteins
We conducted MWAS to test for pQTM associations between 772,619 CpG sites and 4058 circulating protein levels (corresponding to 4235 SOMAmer measurements; Supplementary Data 1). The MWAS population included 774 individuals from Generation Scotland (mean age 60 years [SD 8.8], 56% Female; Supplementary Data 2). 143 principal components explained 80% of the cumulative variance in the 4235 measurements (Supplementary Fig. 1 and Supplementary Data 3). A threshold for multiple testing based on these components was applied across all MWAS ($P < 0.05/(143 \times 772,619) = 4.5 \times 10^{-10}$).

In our basic model adjusting for age, sex and available genetic pQTL effects from Sun et al.[20] 238,245 pQTMs (2107 *cis* and 236,138 *trans*, representing 0.005% of tested associations) had $P < 4.5 \times 10^{-10}$ (Supplementary Data 4). In our second model that further adjusted for Houseman-estimated white blood cell proportions[21], there were 3,213 associations (453 *cis* and 2760 *trans*) that had $P < 4.5 \times 10^{-10}$ (Supplementary Data 5). Smoking status and BMI are known to have well-characterised DNAm signatures[22,23]; fully-adjusted models were

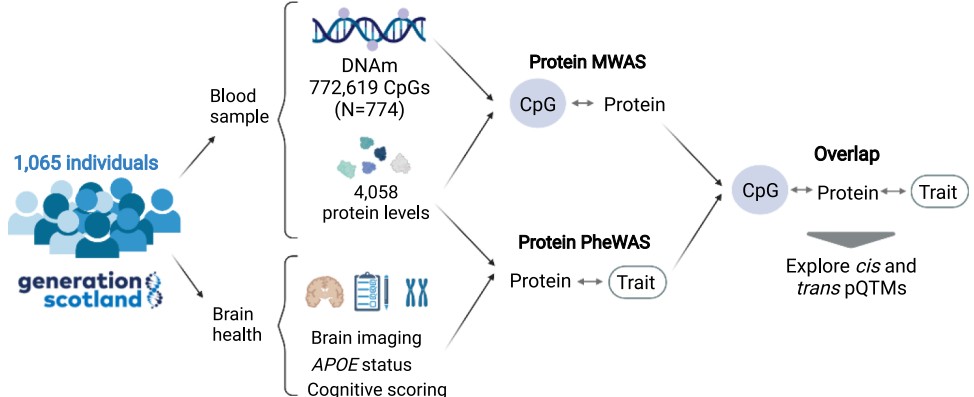

**Fig. 1 | Methylome and phenome study of the plasma proteome in relation to brain health study design.** A total of 4058 plasma proteins (corresponding to 4235 SOMAmers) were measured in 1065 individuals in Generation Scotland. A methylome-wide association study (MWAS) of each plasma protein level was conducted in 774 individuals that represented a nested subset of the full sample that had DNAm measurements available. A phenome-wide protein association study (Protein PheWAS) identified protein levels that were associated with a minimum of one brain health trait ($N \geq 909$). Overlapping the protein MWAS and PheWAS results identified pQTMs that involved protein markers of brain health. The functional roles of proteins and CpGs involved in this subset were explored further, with approaches tailored to interpretation of *cis* and *trans* pQTMs. Created with BioRender.com.

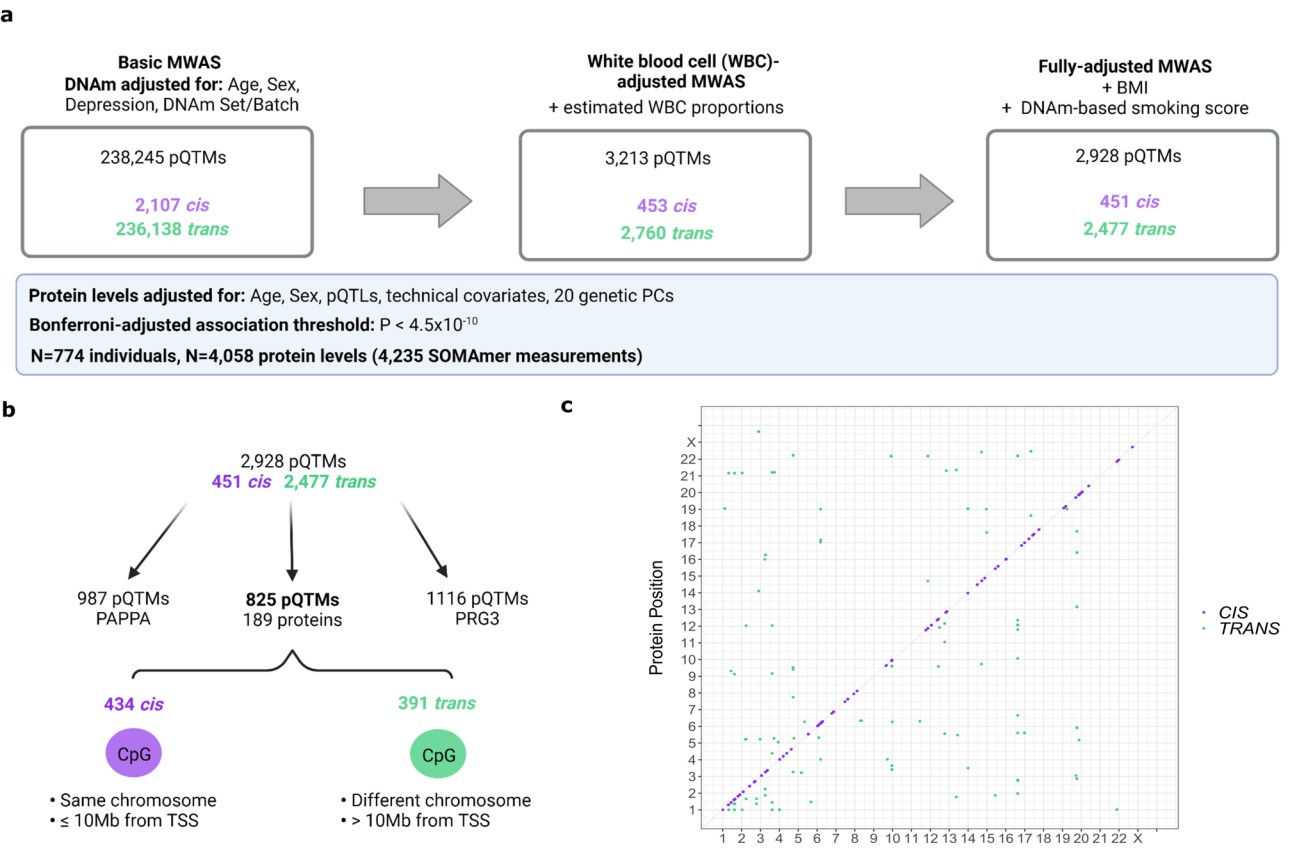

**Fig. 2 | Methylome-wide association studies (MWAS) of 4058 plasma proteins.**
**a** Summary of MWAS results for 4058 protein levels in Generation Scotland
(*N* = 774). The number of protein quantitative trait methylation loci (pQTMs) that
had $P < 4.5 \times 10^{-10}$ (Bonferroni threshold for multiple testing adjustment) in the
basic, white blood cell proportion (WBC)-adjusted and fully-adjusted models. *Cis*
associations (purple) and *trans* associations (green) are summarised for each
model. Covariates used to adjust DNAm are described for each model. Normalised
protein levels were adjusted for age, sex, 20 genetic principal components (PCs),
protein quantitative trait loci (pQTLs) and technical variables and scaled to have a
mean of 0 and standard deviation of 1. Results were generated through linear
regression models. Full summary statistics with full *P* values can be accessed in
Supplementary Data 6. Created with BioRender.com. **b** Flow diagram showing the
distinction between the highly pleiotropic PAPPA and PRG3 protein pQTMs and the
825 pQTMs that involved the levels of a further 189 proteins. TSS: transcriptional
start site of the protein gene. The 434 *cis* pQTMs (purple) lay on the same chro-
mosome and ≤ 10 Mb from the transcriptional start site (TSS) of the protein gene,
whereas the 391 *trans* pQTMs (green) lay >10 Mb from the TSS of the protein gene
or on a different chromosome. Created with BioRender.com. **c** Genomic locations
for 825 of the 2928 fully-adjusted pQTMs, excluding highly pleotropic associations
for PAPPA and PRG3 protein levels, with cis pQTMs in purple and *trans* pQTMs in
green. Chromosomal location of CpG sites (*x*-axis) and protein genes (*y*-axis) are
presented. A list of the full association counts for each protein and CpG site can be
found in Supplementary Data 8, 9.

therefore further adjusted for these factors. There were 2,928 asso-
ciations (451 *cis* and 2477 *trans*) in the fully-adjusted models (Supple-
mentary Data 6). 2847 pQTM associations were significant in all
models. There were 191 unique proteins with associations in the fully-
adjusted models, corresponding to 195 SOMAmer measurements (two
SOMAmers were present for CLEC11A, GOLM1, ICAM5 and LRP11).
Figure 2 summarises these findings. Genomic inflation statistics for
these 195 SOMAmer measurements (fully-adjusted MWAS) are pre-
sented in Supplementary Data 7. In a sensitivity analysis, restriction of
the threshold for *cis* pQTMs from 10 Mb to 1 Mb from the transcription
start site of the gene encoding the protein yielded 409 cis pQTMs (a
reduction of 42 pQTMs) in the fully-adjusted MWAS. A summary of
known pQTLs[20] and a record of whether these were available for
adjustment is provided in Supplementary Data 8. Characterising the
genomic location of the findings, 46% of *cis* and 29% of *trans* pQTMs in
the fully-adjusted MWAS involved CpGs positioned in either a CpG
Island, Shore or Shelf (Supplementary Data 6).

**Pleiotropic pQTM associations in the fully-adjusted MWAS**
Pleiotropy was observed for both CpG sites and protein levels (Fig. 3).
Nineteen proteins had 10 or more pQTMs in the fully-adjusted MWAS

(Supplementary Data 9). Of the 2928 pQTMs in the fully-adjusted
MWAS, 987 involved Pappalysin-1 (PAPPA) and there were a further
1116 pQTMs that involved the Proteoglycan 3 Precursor protein. The
remaining 825 pQTMs involved 189 unique protein levels, with 434 *cis*
and 391 *trans* associations (Fig. 2). Principal components analyses
indicated high correlations between CpGs associated with the pleio-
tropic proteins PAPPA and PRG3, whereas the CpGs involved in the
remaining 825 pQTMs were largely uncorrelated (Supplementary
Fig. 2). pQTM frequencies for the 1837 unique CpGs selected in the
fully-adjusted models, with their respective genes and EWAS catalog[24]
lookup of epigenome-wide significant ($P < 3.6 \times 10^{-8}$) phenotypic
associations is presented in Supplementary Data 10. Of these CpGs,
sites within the *NLRC5, SLC7A11* and *PARP9* gene regions exhibited the
highest levels of pleiotropy (Fig. 3).

The pleiotropic findings for PAPPA and cg07839457 (*NLRC5* gene)
replicated previous MWAS results from Zaghlool et al.[14] (944 indivi-
duals, with 1123 protein SOMAmers). Of the 98 pQTMs identified by
Zaghlool et al., 81 were comparable (both the protein and CpG sites
from the 98 pQTMs were available across both MWAS). Of these 81
pQTMs, 26 replicated at our significance threshold ($P < 4.5 \times 10^{-10}$) with
the same direction of effect, a further 16 replicated at the epigenome-

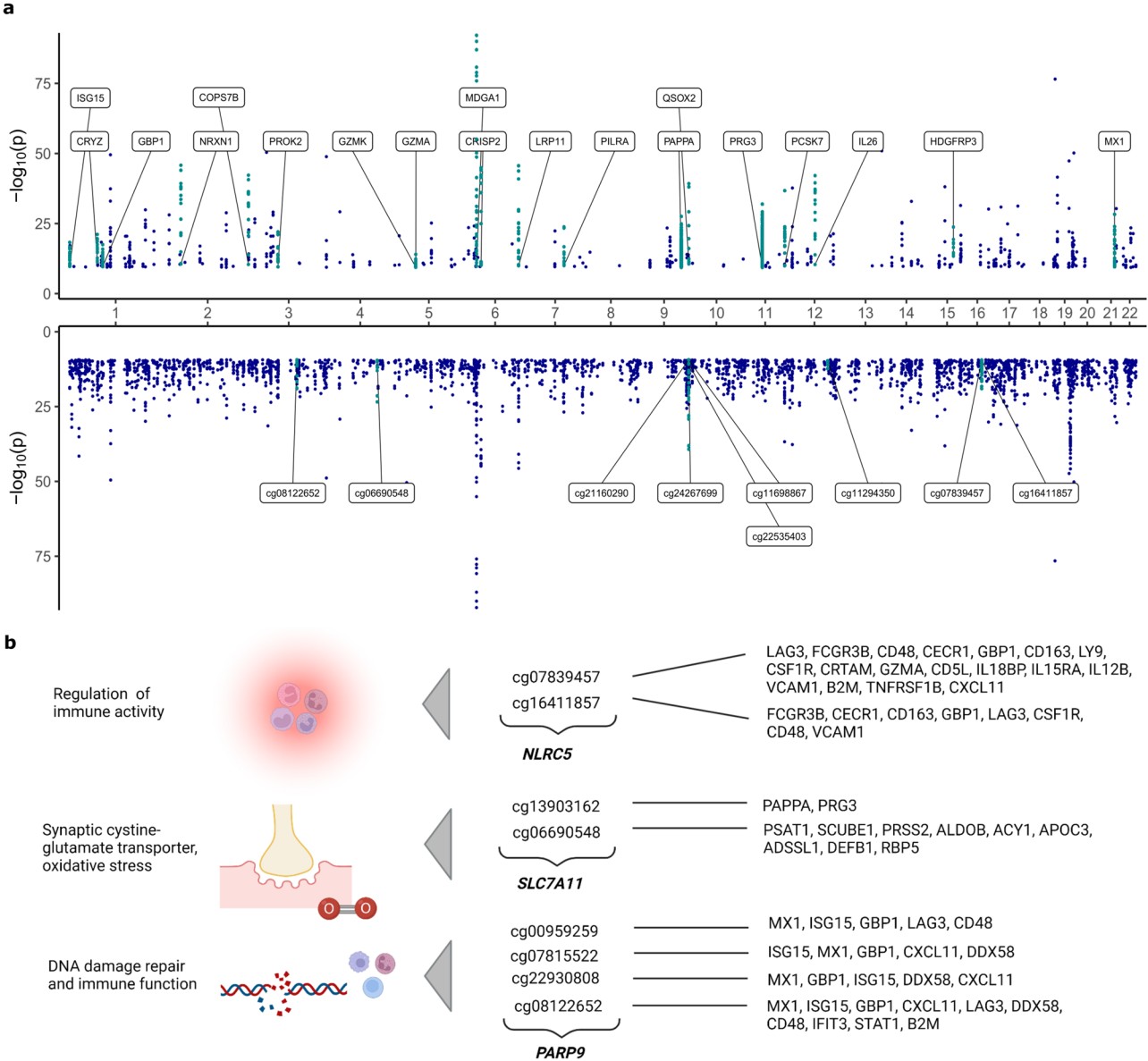

**Fig. 3 | Pleiotropic associations in the fully-adjusted methylome-wide association studies (MWAS). a** pQTMs that had $P < 4.5 \times 10^{-10}$ (Bonferroni threshold for multiple testing adjustment) in the fully-adjusted MWAS are plotted as individual points (dark blue) with chromosomal locations of the 191 protein genes (upper) and the 1837 CpGs (lower) on the x-axis. 19 proteins with ≥10 associations with CpGs are highlighted in turquoise and labelled on the upper plot. Nine CpGs with ≥6 associations with protein levels are highlighted in turquoise on the lower plot. Results were generated through linear regression models. Full summary statistics with full P values can be accessed in Supplementary Data 6. **b** A selection of CpGs with highly pleiotropic signals in the fully-adjusted MWAS and the corresponding function of the gene the CpGs were located within. Created with BioRender.com.

wide significance threshold ($P < 3.6 \times 10^{-8}$)[25] and a further 39 replicated at nominal $P < 0.05$ (Supplementary Data 11 and Supplementary Fig. 3). When accounting for 26 pQTMs that were previously reported by Zaghlool et al. and 10 pQTMs that were previously reported by Hillary et al.[14,19], 2892 of the 2928 fully-adjusted pQTMs were previously unreported. Of these 2892 pQTMs, 1109 involved the levels of 41 proteins that were measured by Zaghlool et al. (973 pQTMs for PAPPA and 136 additional pQTMs for the levels of 40 proteins), whereas 1783 pQTMs involved the levels of proteins that were previously unmeasured (1116 pQTMs for PRG3 and 667 further pQTMs for 148 proteins).

## Proteome associations with brain health phenotypes

We next conducted a proteome-wide association study of brain health characteristics (protein PheWAS of brain imaging, cognitive scoring and *APOE* e4 status, alongside age and sex; Fig. 4). Distribution plots for the seven cognitive scores and seven brain imaging phenotypes are

presented in Supplementary Figs. 4, 5. A maximum sample of 1065 individuals was available (mean age 59.9 years [SD 9.6], 59% Female; Supplementary Data 2); all 774 individuals from the pQTM study were included in these analyses. A threshold for multiple testing adjustment was calculated based on 143 independent components that explained >80% of the 4235 SOMAmer levels (Supplementary Data 3 and Supplementary Fig. 1). This equated to $P < 0.05/(143) = 3.5 \times 10^{-4}$. The levels of 587 plasma proteins were associated with age and 545 were associated with sex, with 222 proteins common to both phenotypes (Supplementary Data 12). When comparable associations from three studies (with $N > 1000$) were tested[20,26,27], 97% of age and 98% of sex associations replicated in one or more of studies (Supplementary Data 12).

There were 191 unique protein markers that had a total of 405 associations with brain health characteristics (Supplementary Fig. 6 and Fig. 4a). These consisted of 95 brain imaging (Supplementary

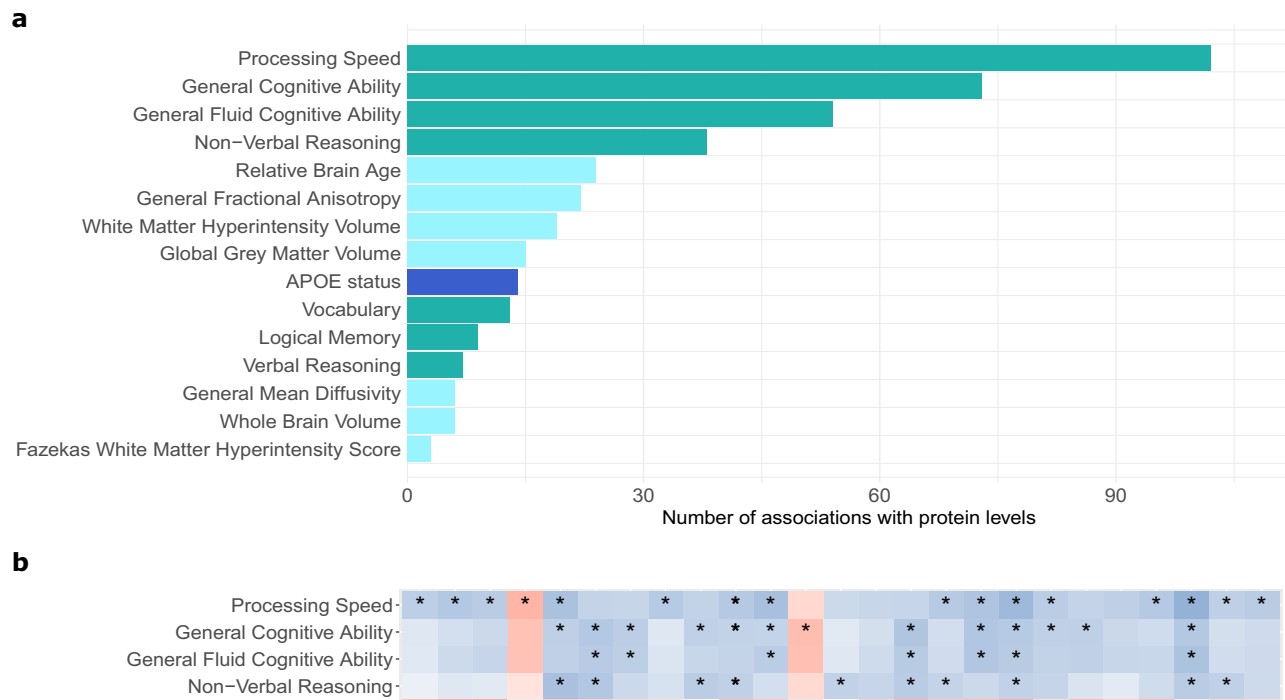

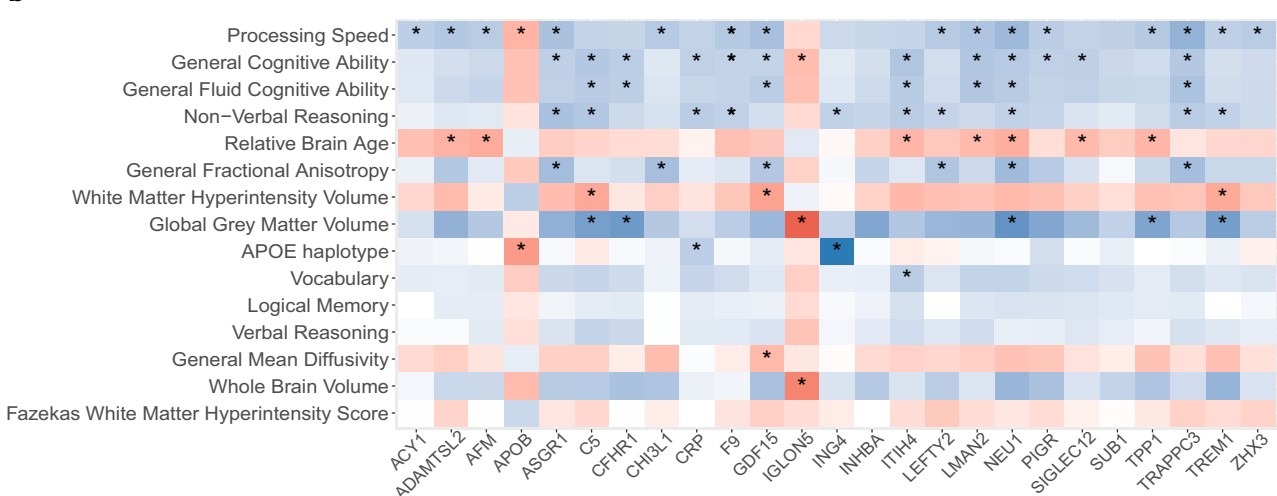

**Fig. 4 | Phenome-wide associations studies (PheWAS) of 4058 plasma proteins and brain health. a** Number of protein marker associations with $P < 3.5 \times 10^{-4}$ for each of the 15 traits related to brain health in the phenome-wide protein association studies (protein PheWAS). These studies included a maximum sample of 1065 individuals with protein measurements from Generation Scotland and tested for associations between 15 phenotypes and the levels of 4058 plasma proteins via linear mixed effects regression. Cognitive score (green), brain imaging (light blue) and *APOE* e4 status (dark blue) associations are summarised. Full summary statistics for the 405 associations with *P* values are presented in Supplementary Data 17. All associations were generated through linear regression and were adjusted for multiple testing correction. **b** Heatmap of standardised beta coefficients for 77 of the 405 protein PheWAS associations ($P < 3.5 \times 10^{-4}$ indicated by an asterisk). These include three proteins that had associations with both *APOE* e4 status and one or more cognitive scores, in addition to 22 proteins that had associations with both a brain imaging measure and a cognitive score. Negative and positive beta coefficients are shown in blue and red, respectively. A heatmap describing the full 405 associations for *APOE* e4 status, cognitive scores and brain imaging measures is available in Supplementary Fig. 6. All associations were generated through linear regression and were adjusted for multiple testing correction.

Data 13), 296 cognitive test score (Supplementary Data 14) and 14 *APOE* e4 status (Supplementary Data 15) associations. Supplementary Data 16 stratifies these associations by direction of effect and Supplementary Data 17 provides full summary statistics for all 405 associations. Of the seven brain morphology traits, Relative Brain Age and General Fractional Anisotropy (gFA) had the largest number of associations, with 24 and 22 protein markers identified, respectively. Of the cognitive score traits, Processing Speed and General Cognitive Ability scores were associated with the highest number of protein markers (102 and 73, respectively). The underlying data for the 14 *APOE* e4 status associations are plotted in Supplementary Fig. 7.

Stratifying the 405 associations by direction of effect revealed that the majority (89%) of associations involved higher levels of the proteins that were associated with less favourable brain health (Supplementary Data 16). Eighty-seven of the 405 associations involved protein levels that were associated with more favourable brain health;

this signature included the levels of SLITRK1, NCAN and COL11A2. Higher levels of ASB9, RBL2, HEXB and SMPD1 were associated with poorer brain health. Protein interaction network analyses for the genes corresponding to the 191 protein markers (Supplementary Fig. 8) indicated that many of the proteins clustered together, implying shared underlying functions. An inflammatory cluster including CRP, ITIH4, C3, C5, COL11A2 and SIGLEC2 was present and higher levels of these markers were associated with poorer brain health outcomes. Gene set enrichment analyses on the 191 genes corresponding to the protein markers (Supplementary Fig. 9) supported the link between many of the proteins associated with poorer brain health and the innate immune system, while also implicating extracellular matrix, lysosomal, metabolic and additional inflammatory pathways. Tissue expression profiles of the 191 genes (Supplementary Fig. 10) indicated that many of the markers were expressed non-neurological tissues; however, some proteins were expressed in nervous tissues. Markers

such as ASB9 and NCAN were found to be consistently identified across multiple brain imaging traits as markers of poorer and better brain health, respectively (Supplementary Data 16). While many of the associations for brain imaging measures identified proteins that were distinct from those found for cognitive scores and *APOE* e4 status, 22 protein markers were associated with both a cognitive score and a brain imaging trait (Fig. 4b and Supplementary Data 18). A principal components analysis of the 22 protein levels was conducted. The first five components had an eigenvalue >1 and a cumulative variance of >80% was explained by the first 10 components. These are both commonly-used thresholds for deciding how many principal components to retain[28] (Supplementary Fig. 11). Three *APOE* e4 status markers (ING4, APOB and CRP) were also associated with cognitive scores (Fig. 4b).

### Replication of protein PheWAS associations
Six of the 14 *APOE* e4 status associations replicated previous SOMAmer protein findings (N SOMAmers = 4785 and *N* participants = 227)[10], and eight previously unreported relationships involved NEFL, ING4, PAF, MENT, TMCC3, CRP, FAM20A and PEF1. Several of the markers for cognitive function were identified in previous work relating Olink proteins to cognitive function (such as CPM)[29] and work that characterised SOMAmer signatures of cognitive decline and incident Alzheimer's disease (such as SVEP1)[8]. No studies have performed SOMAmer-based, whole proteome PheWAS studies of the brain imaging and cognitive score traits we have profiled in a heathy ageing population that were not enriched for neurodegenerative diseases. However, replication of associations from several studies[9,29,30] was found for a small subset of associations (Supplementary Data 19).

### Integration of the brain health proteome with our pQTM dataset
Differential DNAm signatures were explored for the 191 protein markers that had $P < 3.5 \times 10^{-4}$ in associations with either cognitive scores, brain imaging measures or *APOE* e4 status in the protein PheWAS. Of the 191 proteins, 17 had pQTMs in the fully-adjusted MWAS. Higher levels of 15 of these proteins were associated with poorer brain health, while AMY2A and CST5 were associated with more favourable brain health. There were a total of 35 pQTMs involving 31 unique CpGs that were located within 20 distinct genes (Supplementary Data 20), with 15 *trans* (Fig. 5) and 20 *cis* associations. All pQTMs were previously unreported. The 20 *cis* pQTMs involved the levels of CHI3L1, IL18R1, SIGLEC5, OLFM2, UGDH, CRHBP, AMY2A and CFHR1 proteins. The *trans* pQTMs involved the levels of SCUBE1, RBL2, TNFRSF1B, CST5, HEXB, ACY1, CRTAM, SMPD1 and RBP5 proteins.

Of the 20 cis pQTMs, 11 involved CpGs in different genes to the protein-coding gene on the same chromosome, whereas the remaining 9 pQTMs involved CpGs located within the protein-coding gene. Several CpG sites were associated with multiple protein levels in the *trans* pQTMs (Fig. 5). DNAm at site cg06690548 in the *SLC7A11* gene was associated with RBP5, ACY1 and SCUBE1 levels. The cg11294350 site in the *CHPT1* gene was associated with HEXB and SMPD1 levels. The cg07839457 site in the *NLRC5* gene was associated with the levels of CRTAM and TNFRSF1B. There was also a protein that had several *trans* associations with multiple CpG sites; pQTMs were identified between circulating RBL2 levels and cg01132052, cg0539861, cg18487916, cg27294008 and cg18404041, within the *NEK4/ITIH3/ITIH1* gene region of chromosome 3.

### Functional mapping of neurological pQTMs
A lookup that integrated information from the GoDMC and eQTLGen databases assessed whether pQTMs were partially driven by an underlying genetic component. This identified methylation quantitative trait loci (mQTLs) for CpGs that were associated with CHI3L1, IL18R1 and SIGLEC5 levels and were also expression quantitative trait loci (eQTLs) for the respective proteins (Supplementary Data 20). Further visual inspection of the distributions for the 35 pQTMs indicated that trimodal distributions – suggestive of unaccounted SNP effects−were present for CpGs involved in seven of the pQTMs (Supplementary Fig. 12).

Tissue expression profiles for the 33 genes that were linked to either CpGs or proteins in the 35 neurological pQTMs are summarised in Supplementary Fig. 13. Gene set enrichment for these 33 genes identified enrichment for immune effector pathways in a subset of 11 genes, whereas a cluster of four genes (*SMPD1, HEXB, AMY2A* and *AMY2B*) were enriched for amylase and hydrolase activity (Supplementary Fig. 14).

Of the 35 pQTMs, seven had CpGs that were located in either a CpG Shore or Shelf position and there were 13 that were located either 1500 bp or 200 bp from the TSS of the protein-coding gene (Supplementary Data 20). Fifteen pQTMs involved CpGs that were located in the gene body and 7 were located in either the first exon or UTR regions (Supplementary Data 20).

Promoter-capture Hi-C and ChIP-sequencing integration were used to assess the interactions and chromatin states of our pQTMs and associated CpG loci. This analysis focused on 11 of the 20 *cis* pQTMs that involved CpGs on the same chromosome as the protein-coding gene, but was located in a different gene. Mapping information is presented for the seven proteins involved in these pQTMs in Supplementary Figs. 15−21. In all instances, we found evidence of spatial co-localisation of these genes using promoter-capture Hi-C data from brain hippocampal tissue. We attempted to contextualise these sites further with ChIP-seq (ENCODE project) analyses of active chromatin marks H3K27ac and H3K4me1 and repressive chromatin H3K4me3 and H3K27me3 in both peripheral blood mononuclear cells (PBMCs) and brain hippocampus. ChIP-seq data suggested that in many instances there were shared regulatory regions that existed across both blood and hippocampal samples that were hubs for local promoter interactions. For example, promoter loops were found linking the *S100Z* and *CRHBP* genes, with a signature of activating (H3Kme1 and H3K27ac) and silencing (H3k27me and H3K4me3) marks (normally considered bivalent chromatin) that may form the basis for shared regulation of this gene locus.

## Discussion
We have conducted a large-scale integration of the circulating proteome with indicators of brain health and blood-based DNA methylation. We characterised 191 protein markers that were associated with either brain imaging measures, cognitive scores or *APOE* e4 status in an ageing population. We also report methylome-wide characterisations for the SOMAscan® panel V.4 (4058 protein measurements) in a nested subset of this population. By overlapping these datasets, we uncovered 35 methylation signatures for 17 protein markers of brain health. We delineated pQTM CpGs that had evidence of underlying genetic influence and characterised the potential for chromatin interactions for genes involved in *cis* pQTMs. As this population consists of older individuals that were not enriched for neurodegenerative diseases, the markers we identify are likely indicators of healthy brain ageing.

Many of the 191 proteins identified in the protein PheWAS were part of inflammatory clusters with shared functions in acute phase response, complement cascade activity, innate immune activity and cytokine pathways. Tissue expression analyses suggested that a large proportion of the 191 protein markers were not expressed in the brain; this supports work suggesting that sustained peripheral inflammation influences general brain health[31,32] and accelerates cognitive decline[17,33–35]. However, a subset of proteins were expressed in the central nervous system. Given that leakage at the blood-brain-barrier interface has been hallmarked as a part of healthy brain ageing[36,37], there is a possibility that brain-derived proteins may enter the bloodstream as biomarkers. SLIT and NTRK Like Family Member 1 (SLITRK1),

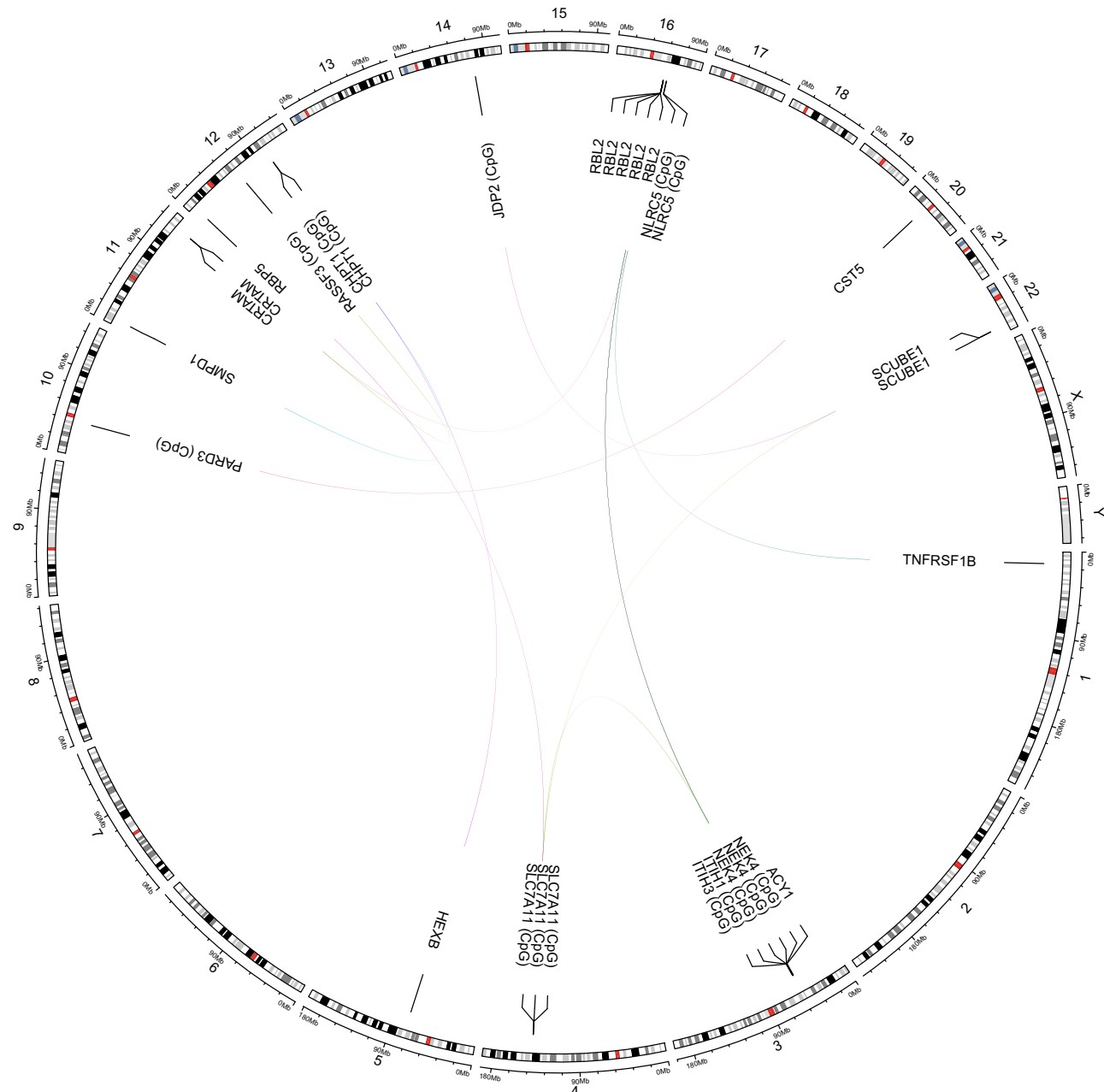

**Fig. 5 | *Trans* pQTMs involving protein markers of brain health.** Circular plot showing 15 *trans* pQTM associations between DNAm at 11 CpG sites and the levels of nine protein markers of brain health that had $P < 4.5 \times 10^{-10}$ (Bonferroni threshold for multiple testing adjustment) in the fully-adjusted MWAS. Chromosomal positions are given on the outermost circle. Details of the full set of 35 pQTMs for protein markers of brain health are provided in Supplementary Data 20 with $P$ values. Results were generated through linear regression models.

Neurocan (NCAN) and IgLON family member 5 (IGLON5) were examples of proteins expressed in brain for which higher levels associated with either larger grey matter volume, larger whole brain volume, or higher general fractional anisotropy. SLITRK1 localises at excitatory synapses and regulates synapse formation in hippocampal neurons[38]. NCAN is a component of neuronal extracellular matrix and is linked to neurite growth[39]. IGLON5 has been implicated in maintenance of blood−brain−barrier integrity and an anti-IGLON5 antibody disease involves the deterioration of cognitive health[40]. Taken together, the protein markers identified in the PheWAS may, therefore, reflect pathways that could be targeted to improve brain health.

Integration of our fully-adjusted protein MWAS dataset revealed 35 associations between DNAm and 17 protein markers of brain health (Fig. 6; Supplementary Data 20). All 35 associations were previously unreported. While this study is focused on blood DNAm−limiting generalisation to brain DNAm−many of the 35 pQTMs involved CpGs and proteins that have been previously implicated in neurological processes. DNAm at site cg06690548 (located in the *SLC7A11* gene) was of particular interest; differential DNAm at this CpG in blood has been identified as a causal candidate for Parkinson's disease ($N > 900$ cases and $N > 900$ controls)[41]. Xc- is the cystine-glutamate antiporter encoded by *SLC7A11*, which facilitates glutamatergic transmission, oxidative stress defence and microglial response in the brain[42,43] and is a target for the neurodegeneration-associated environmental neurotoxin β-methylamino-L-alanine[41]. Analyses in the wider Generation Scotland cohort suggests that cg06690548 is a site associated with alcohol consumption[44]. The proteins associated with cg06690548 in the subset of this cohort that we assessed (ACY1, SCUBE1 and RBP5)

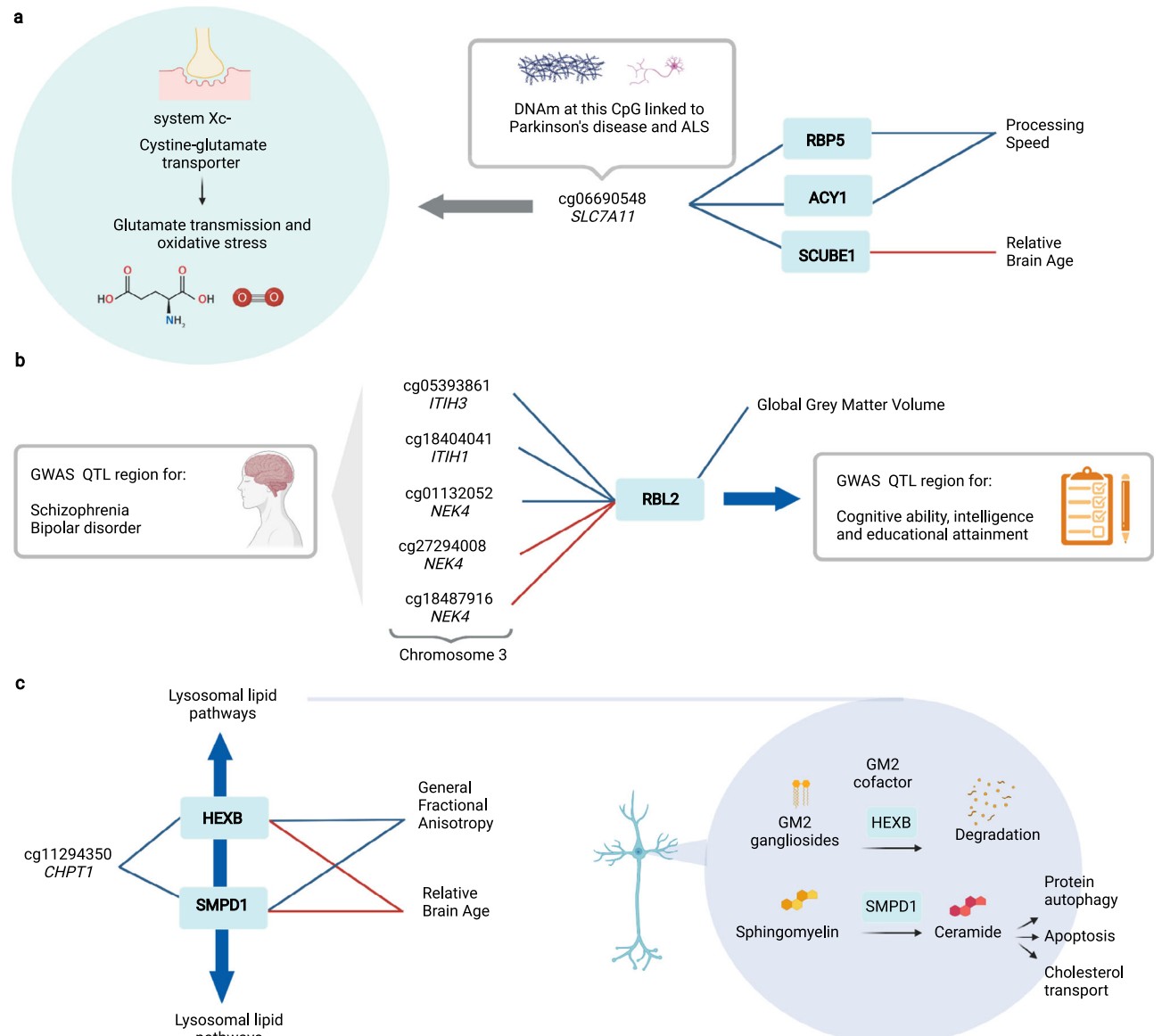

**Fig. 6 | Exploration of *trans* pQTMs for protein markers of brain health. a** Three *trans* associations with the CpG site cg06690548 in the *SLC7A11* gene, which encodes a synaptic protein that is involved in glutamate transmission and oxidative stress. cg06690548 has been implicated in methylome-wide studies of Parkinson's disease and Amyotrophic lateral sclerosis (ALS) risk. **b** Five *trans* associations between CpGs in the *ITIH3/ITIH1/NEK4* region on chromosome 3 and the levels of RBL2, which was associated with lower Global Grey Matter Volume. The *RBL2* gene has been implicated in genome-wide associations studies (GWAS) of cognitive ability, intelligence and educational attainment. The *ITIH3/ITIH1/NEK4* region has been implicated in GWAS of Schizophrenia and Bipolar disorder. **c** Two *trans* associations between DNAm at cg11294350 in the *CHPT1* gene and two proteins with lysosomal-associated function (SMPD1 and HEXB) that were associated with higher Relative Brain Age and lower General Fractional Anisotropy. Associations with a positive beta coefficient are denoted as red connecting lines, whereas associations with a negative beta coefficient are denoted as blue connecting lines. The full 35 pQTMs for protein markers of brain health (15 *trans* and 20 *cis*) can be found in Supplementary Data 20. All associations were generated using linear regression and were adjusted for multiple testing. Created with BioRender.com.

have known links to liver function[45–47]. DNAm at cg06690548 in blood has also been recently implicated in the largest MWAS of amyotrophic lateral sclerosis (ALS) to date (6763 cases, 2943 controls)[48]. Given that ACY1, SCUBE1 and RBP5 were markers for either lower processing speed and higher relative brain age, the CpG sites we identify in this study−such as cg06690548−may be important plasma markers for mediation of environmental risk on brain health that merit further exploration. cg06690548 lies within the first intron of *SLC7A11*[41], indicating that this site is of potential functional significance.

The presence of *NLRC5*-associated CpGs and various other inflammatory proteins in our neurological protein pQTMs suggests that the methylome may capture an inflammatory component of brain health. Many of the genes corresponding to CpGs and proteins involved in the 35 pQTMs were enriched for immune effector processes and were not expressed in brain. However, some genes did show evidence for brain-specific expression, such as acid sphingo-myelinase (*SMPD1*) and Hexosaminidase Subunit Beta (*HEXB*). The HEXB and SMPD1 proteins associated with DNAm at cg11294350 (in the *CHPT1* gene), are involved in neuronal lipid degradation in the brain and have been associated with the onset of a range of neurodegen-erative conditions[49–52]. RBL2 is another protein that had partial expression signals across brain regions; the *NEK4/ITIH3/ITIH1* region was the location for five CpGs with differential DNAm linked to RBL2 levels. This region is implicated in schizophrenia and bipolar disorder by several large-scale, genome-wide association studies (GWAS)[53–56]. Similarly, the *RBL2* locus has been associated with intelligence,

cognitive function and educational attainment in GWAS ($n > 260,000$ individuals)[34,57,58].

Given that this study utilised CpGs from the Illumina EPIC array, 15 of the 31 unique CpGs did not have mQTL characterisations in public databases, which primarily comprise results from the earlier 450 K array. However, our plots showing pQTM associations suggested that for several CpGs (such as cg11294350 that associated with SMPD1 and HEXB), there may be a partial genetic component influencing DNAm. As mQTLs tend to explain 15–17% of the additive genetic variance of DNAm[59], it is possible that the signals we isolate in these instances are partially driven by genetic loci, but are also likely driven by unmeasured environmental and biological influences. In the case of SIGLEC5, IL18R1 and CHI3L, mQTLs were identified that were also eQTLs, providing evidence that mQTLs for these CpG sites were possible regulators of protein expression.

Integration of promoter-capture Hi-C chromatin interaction and ChIP-seq databases[60] provided evidence for long-range interaction relationships for *cis* pQTMs with CpGs in different gene regions that are proximal to the protein-coding gene of interest. This suggests that in such instances, the pQTMs may reflect regulatory relationships in the 3-dimensional genomic neighbourhood. The pQTMs therefore direct us towards pathways that can be tested in experimental constructs. Positional information suggested that many CpGs involved in neurological pQTMs lay within 1500 bp of the TSS of the respective protein-coding gene. While positional information of CpGs is thought to infer whether DNAm is likely to play a role in the expression regulation of nearby genes, this is still somewhat disputed. Some studies suggest that transcription factors regulate DNAm[61] and differential methylation at gene body locations predicts dosage of functional genes[62]. Additionally, the DNAm signatures of proteins we quantify represent widespread differences across blood cells that are related to circulating protein levels and are therefore not derived from the same cell-types as proteins. Despite this limitation, previous work supports DNAm scores for proteins as useful markers of brain health, suggesting there is merit in integrating DNAm signatures of protein levels in disease stratification[18].

Our study has several limitations. First, though full replication of our results was not possible, our replication of pQTMs identified by Zaghlool et al.[14] reinforces inflammation signalling as intrinsic to the methylome signature of blood proteins. This also suggests that pQTMs may be common across ancestries. Second, we observed a substantial inflation for PAPPA and PRG3 proteins. While comprehensive adjustment for estimated immune cells was performed and the remainder of CpGs involved in pQTMs did not show high correlations (Supplementary Fig. 2), concurrently measured blood components such as haemoglobin, red blood cells and platelets were not available. Future studies should seek to resolve signals with more detailed blood-cell phenotyping and immune cell estimates[63]. Third, 89% of the proteins identified in our protein PheWAS did not have epigenetic pQTMs; this may be due to 1) the presence of pathways relating to neurological disease that are not reflected by blood immune cell DNAm, 2) underpowered analyses, or 3) the presence of indirect associations that are not captured by our MWAS approach. Fourth, the extent of non-specific and cross-aptamer binding with SOMAmer technology has not been fully resolved[64]. Fifth, there are likely unknown genetic influences on pQTMs. Further characterisation of pQTLs and advances in multi-omic modelling techniques[15] will aid in the separation of genetic and environmental influences on epigenetic signatures. Sixth, differences in blood and brain DNAm and pQTLs are emerging; these indicate that blood-based markers may not fully align to biology of brain degeneration[65,66]. However, our ChIP-seq analysis of chromatin regulation suggested that some regulatory states may persist between blood and brain. Seventh, profiling DNAm signatures alone cannot capture the full role of the epigenome in brain health. Integration of more diverse epigenetic markers will be critical to further resolve these

relationships. Finally, though we have incorporated a wide portfolio of brain health measures, we recognise that these are not extensive. Increasing triangulation across modalities, as we have shown here, will be useful in identifying candidate markers.

In conclusion, by integrating epigenetic and proteomic data with cognitive scoring, brain morphology and *APOE* e4 status, we identify 191 protein markers of brain health. We characterise DNAm signatures for all 4058 proteins included in the study, uncovering 35 associations between differential DNAm and the levels of 17 of the protein markers of brain health. These data identify candidate targets for the preservation of brain health and may inform risk stratification approaches.

## Methods

### The Generation Scotland sample population
A YouTube video providing an overview of this study and detailing how summary statistics can be accessed is available at: https://www.youtube.com/channel/UCxQrFFTIltF25YKfJTXuumQ. The Stratifying Resilience and Depression Longitudinally (STRADL) cohort used in this study is a subset of $N = 1188$ individuals from Generation Scotland: The Scottish Family Health Study (GS). Generation Scotland constitutes a large, family-structured, population-based cohort of >24,000 individuals from Scotland[67]. Individuals were recruited to GS between 2006 and 2011. During a clinical visit detailed health, cognitive, and lifestyle information was collected in addition to biological samples. Of the 21,525 individuals contacted for participation, $N = 1188$ completed additional health assessments and biological sampling -5 years after GS baseline[68]. Of these, $N = 1,065$ individuals had proteomic data available and $N = 778$ of these had DNAm data available. Four individuals from this subset were excluded from the DNAm sample due to having incomplete depression status information, leaving 774 individuals available for analyses. Supplementary Data 2 summarises the demographic characteristics across the two groups, with descriptive statistics for phenotypes.

### Proteomic measurement
SOMAscan® V.4 technology was used to quantify plasma protein levels. This aptamer-based assay facilitates the simultaneous measurements of multiple Slow Off-rate Modified Aptamers (SOMAmers)[69]. SOMAmers were processed for 1065 individuals from the STRADL subset of Generation Scotland. Briefly, binding between plasma samples and target SOMAmers was achieved during incubation and quantification was recorded using a fluorescent signal on microarrays. Quality control steps included hybridisation normalisation, signal calibration and median signal normalisation to control for inter-plate variation. Full details of quality control stages are provided in Supplementary Information. In the final dataset, 4235 SOMAmer epitope measures were available in 1065 individuals and these corresponded to 4058 unique proteins (classified by common Entrez gene names). Supplementary Data 1 provides annotation information for the 4235 SOMAmer measurements that were available.

### DNAm measurement
Measurements of blood DNAm in the STRADL subset of GS subset were processed in two sets on the Illumina EPIC array using the same methodology as those collected in the wider Generation Scotland cohort[70–72]. Quality control details are provided in Supplementary Information. Briefly, samples were removed if there was a mismatch between DNAm-predicted and genotype-based sex and all non-specific CpG and SNP probes (with allele frequency >5%) were removed from the methylation file. Probes which had a beadcount of less than 3 in more than 5% of samples and/or probes in which >1% of samples had a detection $P > 0.01$ were excluded. After quality control, 793,706 and 773,860 CpG were available in sets 1 and 2, respectively. These sets were truncated to include a total of 772,619

common probes and were joined together for use in the MWAS, with 476 individuals included in set 1 and 298 individuals in set 2. DNAm-specific technical variables (measurement batch and set) were adjusted in all MWAS models.

## Phenotypes in Generation Scotland

All phenotypes in Generation Scotland MWAS and PheWAS samples are summarised in Supplementary Data 2. An epigenetic score for smoking exposure, EpiSmokEr[73] was calculated for all individuals with DNAm. The meffil[74] implementation of the Houseman method was used to calculate estimated white blood cell proportions for Sets 1 and 2. Blood reference panels were sourced from Reinius et al.[75] with accession GSE35069. The blood gse35069 complete panel was used to imputed measures for Monocytes, Natural Killer cells, Bcells, Granulocytes, CD4+T cells and CD8+T cells. Eosinophil and Neutrophil estimates were also sourced through the blood gse35069 panel. Body mass index (body weight in kilograms, divided by squared height in metres) was available for all individuals, alongside depression status (defined using a research version of the Structured Clinical Interview for DSM disorders (SCID) assessment), which was coded as a binary variable of no history of depression (0) or lifetime episode of depression (1). Five individuals did not have depression status information and were excluded from MWAS and PheWAS analyses, where appropriate. *APOE* e4 status was available for 1050 individuals. *APOE* e4 status was coded as a numeric variable (e2e2 = 0, e2e3 = 0, e3e3 = 1, e3e4 = 2, e4e4 = 2). Fifteen e2e4 individuals were excluded due to small sample size.

Scores from five cognitive tests (Supplementary Fig. 4; Supplementary Data 2) measured at the clinic visit for the STRADL subset of GS were considered. Cognitive scores were measured at the baseline clinic visit[68] and full details are provided in Supplementary Information. Briefly, these included the Wechsler Logical Memory Test (maximum possible score of 50), the Wechsler Digit Symbol Substitution Test (maximum possible score of 133), the verbal fluency test (based on the Controlled Oral Word Association task), the Mill Hill Vocabulary test (maximum possible score of 44) and the Matrix Reasoning test (maximum possible score of 15). Outliers were defined as scores >3.5 standard deviations above or below the mean and were removed prior to analysis. The first unrotated principal component combining logical memory, verbal fluency, vocabulary and digit symbol tests was calculated as a measure of general cognitive ability (g). General fluid cognitive ability (gf) was extracted using the same approach, but with the vocabulary test (a crystallised measure of intelligence) excluded from the model. While highly similar to g, the gf score is exclusive to measures such as memory and processing capability that are considered fluid. gf may therefore be of greater relevance for assessing cognitive decline in ageing individuals.

The derived brain volume measures (Supplementary Fig. 5; Supplementary Data 2) were recorded at two sites (Aberdeen and Edinburgh)[68]. Data processing used the resources provided by the Edinburgh Compute and Data Facility (http://www.ecdf.ed.ac.uk/). Brain volume data included total brain volume (ventricle volumes excluded), global grey matter volume, white matter hyperintensity volume and total intracranial volume. Intracranial volume was treated as a covariate to adjust for head size in all tests including brain volume associations. The derived global white matter integrity measures included gFA and global mean diffusivity. The protocols applied to derive the brain volume measures from T1-weighted scans, and white matter integrity measures from diffusion tensor imaging scans were measured at baseline[35,68,76] and full details are provided in Supplementary Information. Brain Age was estimated using the software package brainageR (Version 2.1; https://doi.org/10.5281/zenodo.3476365, available at https://github.com/james-cole/brainageR), which uses machine learning and a large training set to predict age

from whole-brain voxel-wise volumetric data derived from structural T1 images[3]. This estimate was regressed on chronological age to produce a measure of Relative Brain Age (residuals from the linear model). Outliers for all imaging variables were defined as measurements >3.5 standard deviations above or below the mean and were removed prior to analyses.

## Phenome-wide association analyses

Prior to running protein PheWAS analyses, protein levels were transformed by rank-based inverse normalisation and scaled to have a mean of zero and standard deviation of 1. Models were run using the lmekin function in the coxme R package (Version 2.2–16)[77]. This modelling strategy allows for mixed-effects linear model structure with adjustment for relatedness between individuals. Models were run in the maximum sample of 1065 individuals, with the 4235 protein levels as dependent variables and phenotypes as independent variables. Continuous variables were scaled to mean of zero and variance one and missing data were excluded from lmekin models. Each model adjusted for age and sex (male = 1, reference female = 0). A random intercept was fitted for each individual and a kinship matrix was included as a random effect to adjust for relatedness. Diagnosis of depression (case = 1, reference control = 0) at the STRADL clinic visit was included as a covariate in all models, due to known selection bias for depression phenotypes in STRADL[68]. Clinic study site and protein lag group (storage time before proteomic sequencing) were included as covariates in all models. For the analyses with age and sex as the predictors of interest, two beta coefficients for age and sex were extracted from the same model structure. In the remaining PheWAS models, either numerical *APOE* e4 status variable (e2 = 0, e3 = 1, e4 = 2), cognitive test scores or brain imaging phenotypes were included in addition to the described covariates as scaled predictors. The beta coefficients were extracted for the phenotype in each protein-phenotype association. All analyses of brain volume measures included further adjustment for intracranial volume (ICV) and study site as main effects, in addition to the interaction between these variables. ICV was used to account for head size. Processing batch, and presence or absence of manual intervention during quality control were also included as covariates for volumetric brain imaging associations. The Prcomp function in the stats R package (Version 3.6.2)[78] was used to generate principal components for the 4,235 SOMAmer measurements (N = 1065). 143 components explained >80% of the cumulative variance in protein levels (a commonly-used threshold for the retention of principal components[28]: Supplementary Fig. 1 and Supplementary Data 3). These 143 components were used to derive the PheWAS multiple testing adjustment threshold of $P < 0.05 / 143 = 3.5 \times 10^{-4}$. This method was chosen due to the presence of high intercorrelations within the protein data.

## Epigenome-wide association study of protein levels

Prior to running the MWAS, protein levels for 774 individuals with complete phenotypic information were log transformed and regressed on age, sex, study site, lag group, 20 genetic principal components (generated from multidimensional scaling of genotype data from the Illumina 610-Quadv1 array) and known pQTL effects (from a previous genome-wide association study of 4034 SOMAmers targeting 3622 proteins from Sun et al.)[20]. Residuals from these models were then rank-based inverse normalised and taken forward as protein level data. Methylation data were in M-value format and were pre-adjusted for age, sex, processing batch, methylation set and depression status[73]. A second model further adjusted for estimated white blood cell proportions (Monocytes, CD4+T cells, CD8+T cells, BCells, Natural Killer cells, Granulocytes and Eosinophils). While Neutrophil estimates were available, they were excluded due to high correlation (r > 0.95) with Granulocyte proportions (Supplementary Fig. 22). Finally, the fully-

adjusted model further regressed DNAm onto an epigenetic score for smoking, EpiSmokEr[73] and BMI.

Omics-data-based complex trait analysis (OSCA)[79] Version 0.41 was used to run EWAS analyses. Within OSCA, a genetic relationship matrix (GRM) was constructed for the STRADL population. A threshold of 0.05 was used to identify 120 individuals likely to be related based on their genetic similarity. For this reason, the MOA method was used to calculate associations between individual CpG sites and protein levels, with the addition of the GRM as a random effect to adjust for relatedness between individuals[79]. CpG sites were the dependent variables and the 4235 proteins were the independent variables.

Four fully-adjusted models did not converge (NAGLU, CFHR2, MST1, PILRA) and were excluded. A threshold for multiple testing correction ($P < 4.5 \times 10^{-10}$) was based on 143 independent protein components with cumulative variance >80% (Supplementary Fig. 1 and Supplementary Data 3) ($P < 0.05/(143 \times 772,619)$ CpGs). A more conservative threshold based on total number of SOMAmers was also considered ($P < 0.05/(4235 \times 772,619) = 1.5 \times 10^{-11}$) and is detailed in Supplementary Data 4-6. pQTMs were classified as *cis* if the CpG was on the same chromosome as the protein-coding gene and fell within 10 Mb of the transcriptional start site (TSS) of the protein gene. pQTMs involving a CpG located on a different chromosome to the protein-coding gene, or >10 Mb from the TSS of the protein gene were classed as *trans*.

Circos plots were created with the circlize package (Version 0.4.12)[80]. BioRender.com was used to create Figs. 1, 2, 3 and 6. All analyses were performed in R (Version 4.0)[81].

## Functional mapping and tissue expression analyses

Functional mapping and annotation[82,83] gene set enrichment and tissue expression analyses were conducted for genes corresponding to protein markers that were identified through the PheWAS study, in addition to genes linked to either CpGs or proteins in the neurological pQTM subset. Protein-coding genes were selected as the background set and ensemble v92 was used with a false discovery rate adjusted $P < 0.05$ threshold for gene set testing. For the genes corresponding to protein markers in the PheWAS a minimum overlapping number of genes was set to 3. The STRING[84] database was queried to build a protein interaction network based on all proteins that had associations in the PheWAS. mQTL and eQTL lookups were performed using the GoDMC[59] and eQTLGen databases[85], respectively. UCSC database searches were used to profile the positional information relating to CpGs in the pQTMs.

Although inter-chromosomal chromatin interactions are unlikely to be stable and persistent, seven proteins with *cis* pQTMs involving CpGs located intra-chromosomally to the proximal protein-coding gene were considered for ChIP-seq and promoter-capture Hi-C mapping to interrogate local chromatin interactions and states that might form the basis for co-regulation of these loci. ChIP-seq data from PBMCs and brain hippocampus were selected from the ENCODE project[86], with accession identifiers available in Supplementary Data 21. Processed promoter-capture Hi-C data for brain hippocampal tissue was integrated from Jung et al.[60], and are available at NCBI Geo with accession GSE86189. Data concerning both promoter-promoter interactions and promoter-other interactions were concatenated and all regions subsequently visualised on the WashU epigenome browser[87].

## Ethics declarations

All components of GS received ethical approval from the NHS Tayside Committee on Medical Research Ethics (REC Reference Number: 05/S1401/89). GS has also been granted Research Tissue Bank status by the East of Scotland Research Ethics Service (REC Reference Number: 20/ES/0021), providing generic ethical approval for a wide range of uses within medical research. All participants included in the current study provided informed consent for the use of their data for biomedical research.

## Reporting summary

Further information on research design is available in the Nature Research Reporting Summary linked to this article.

## Data availability

The fully-adjusted MWAS summary statistics for 4231 protein levels generated in this study have been deposited in the MRC-IEU EWAS catalogue[24]. These files are also available through a Zenodo repository at https://doi.org/10.5281/zenodo.6801458[88].

Datasets generated in this study are made available in Supplementary Data files 1–21. The raw data from Generation Scotland are not available due to them containing information that could compromise participant consent and confidentiality. Generation Scotland is run as a Resource for the research community. Requests to use the Resource are made from: Academic collaborators: employees who are party to the Generation Scotland Collaboration Agreement, or researchers or employees of an academic institution or the NHS. Commercial organisations: specific arrangements have been defined to allow commercial organisations to access Generation Scotland resources. Data can be obtained from the data owners. Instructions for accessing Generation Scotland data can be found here: https://www.ed.ac.uk/generation-scotland/for-researchers/access; the GS Access Request Form can be downloaded from this site. Completed request forms must be sent to access@generationscotland.org to be approved by the Generation Scotland Access Committee.

For any further correspondence and material requests please contact Dr Riccardo Marioni at riccardo.marioni@ed.ac.uk. Source data are provided with this paper.

## Code availability

All R code used in this study is available with open access at the following Github repository: https://github.com/DanniGadd/Epigenome-and-phenome-wide-study-of-brain-health-outcomes.

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

## Acknowledgements

This research was funded in whole, or in part, by the Wellcome Trust [104036/Z/14/Z, 108890/Z/15/Z, 221890/Z/20/Z]. For the purpose of open access, the author has applied a CC BY public copyright licence to any Author Accepted Manuscript version arising from this submission. Wellcome Trust 4-year PhD in Translational Neuroscience–training the next generation of basic neuroscientists to embrace clinical research [108890/Z/15/Z]. D.A.G., R.F.H. Wellcome Trust PhD for clinicians, Edinburgh Clinical Academic Track for Veterinary Surgeons. [225442/Z/22/Z]. R.I.M. Alzheimer's Research UK major project grant ARUK-PG2017B–10. D.L.M., R.E.M. Alzheimer's Society major project grant AS-PG-19b-010. R.E.M. Chief Scientist Office of the Scottish Government Health Directorates (CZD/16/6) and the Scottish Funding Council (HR03006). Generation Scotland: D.J.P., A.M.M. Wellcome Trust award 104036/Z/14/Z. D.J.P., A.M.M., A.S., J.M.W. Wellcome Trust award 220857/Z/20/Z. A.M.M. Medical Research Council. MC_PC_17209. A.M.M. National Institutes of Health. R01MH124873. A.M.M. The European Union's Horizon 2020 research and innovation programme under grant agreement No 847776. A.M.M. Medical Research Council [MR/L023784/2]: Dementias Platform UK: L.S., A.N.H. MRC QTL in Health and Disease Programme Grant MC_UU_00007/10 C.H. Medical Research Council Award to the University of Oxford (grant no. MC_PC_17215). L.S. NIHR Biomedical Research Centre at Oxford Health NHS Foundation Trust. L.S. Medical Research Council (MR/R024065/1). S.R.C. US National Institutes of Health (R01AG054628). S.R.C. Sir Henry Dale Fellowship jointly funded by the Wellcome Trust and the Royal Society (Grant Number 221890/Z/20/Z). S.R.C. Joint grant from the Biology and

Biotechnology Research Council and the Economic and Social Research Council (BB/W008793/1). S.R.C. MRC grants MR/S010351/1, MR/W002566/1 and MR/W002388/1. J.D.S. UK Dementia Research Institute which receives its funding from DRI Ltd, funded by the UK Medical Research Council, Alzheimer's Society and Alzheimer's Research UK. J.M.W. Fondation Leducq Transatlantic Network of Excellence for the Study of Perivascular Spaces in Small Vessel Disease, ref no. 16 CVD 05. J.M.W. The Stroke Association/BHF/Alzheimer's Society 'Rates Risks and Routes to Reduce Vascular Dementia' (R4VaD) Priority Programme Award in Vascular Dementia (16 VAD 07). J.M.W. The Row Fogo Charitable Trust; The Row Fogo Centre for Research into Ageing and the Brain; Ref No AD.ROW4.35. BRO-D.FID3668413. J.M.W. The LACunar Intervention Trial-2 (LACI-2) - British Heart Foundation CS/15/5/. J.M.W. Stroke Association SVD-SOS (SA PG 19\100068). J.M.W. Lister Institute of Preventive Medicine - Research Prize Award for Prof. Daniel Smith - Reference 173096 (funded indirectly). A.S. Stroke Association/BHF/Alzheimer's Society Rates Risks and Routes to Reduce Vascular Dementia (R4VaD) Priority Programme Award in Vascular Dementia (16 VAD 07). E.B. The authors acknowledge the work of Rebecca Madden, Marco Squillace and Laura Klinkhamer who aided in the quality control of volumetric brain imaging data.

## Author contributions

D.A.G., and R.E.M., were responsible for the conception and design of the study. D.A.G. carried out the data analyses. D.A.G., and R.E.M., drafted the article. S.R.C., and H.W., advised on methodology. R.F.H., and D.L.M., contributed to methodology and data analyses. R.I.McG., S.M., R.M.W., L.S., D.L.M., R.M.W., A.C., A.N.H., C.H., K.L.E., D.J.P., H.W., A.M.M., and S.R.C., contributed to data methylation and proteomic data collection and preparation. A.S., M.C.B., M.A.H., E.V.B., J.D.S., S.X., C.G., and J.M.W processed the brain imaging data. D.A.O., G.M.T., and C.R., provided scientific counsel. T.C., and N.R., consulted on chromatin analyses. R.E.M., supervised the project. All authors read and approved the final manuscript.

## Competing interests

R.E.M. has received a speaker fee from Illumina and is an advisor to the Epigenetic Clock Development Foundation. A.M.M. has previously received speaker's fees from Illumina and Janssen and research grant funding from The Sackler Trust. R.F.H. has received consultant fees from Illumina. All other authors declare no competing interest.
