## [Peer Review File · Nature Communications]

Integrated methylome and phenome study of the circulating proteome reveals markers pertinent to brain healthREVIEWER COMMENTS

Reviewer #1 (Remarks to the Author):

Gadd et al sought to characterize the associations between plasma proteome with DNA CpG methylation and neurological phenome obtained from the Generation Scotland cohort. Epigenome-wide and phenome-wide studies identified nearly 3k protein quantitative trait methylation loci (pQTM) after multiple testing, and 644 proteins related to various neurological phenotypes, respectively. Integration studies further uncovered 88 epigenetic associations for protein markers of neurological traits with majority of them being novel loci. Overall, this is a straightforward study that may provide some useful resource to the community. However, this study suffers from several significant conceptual and technical shortcomings highlighted below:

Major comments:

1. Methylation on the CpG sites only represents a fraction of "epigenome", other types of epigenetic mechanisms, such as 5-hydroxymethylcytosine, histone modifications and chromatin remodeling have all been implicated to coordinatively involve in neurological diseases. On the other hand, increasing evidence suggests critical epigenetic roles of non-CG methylation in neurological diseases. The present study that solely focuses on CpG methylation could be bias and misleading. The term "epigenetics" or "epigenomics" used throughout the paper is inaccurate and confusing.
2. The authors claimed "As DNAm at CpG sites can affect gene expression in some instances, these relationships may carry information that informs on the regulation of proteins, the biological effector molecules of disease." This is extraordinarily weak as DNA methylation plays differential roles in influencing gene expression at the transcription level, depends on their locations (promoter vs gene bodies, or cis-regulatory elements). The post-transcriptional regulation and translation control are additional layers of regulation. Given the DNA methylation in the plasma is not even from the same population of cells or tissues with circulating proteins, the connection investigated in this study does not represent strong biological insights.
3. The methylated DNA in the plasma could derive from various types of blood cells, secreted vesicles, or any tissues or organs the blood flows. As indicated by the authors, "Blood-based methylation is unlikely to correlate highly with brain methylation in all cases", it is unclear whether the pQTM identified here are specific to neurological diseases showing specific disease manifestations in the brain.
4. For the 255 trans CpG that more than 10Mb from the transcription start site of the protein gene (Fig. 2), the direct transcription regulation is unlikely. Do those CpG sites locate in known cis-regulatory elements to form 3D genomic interaction with these protein targets?
- 5.

Reviewer #3 (Remarks to the Author):

The manuscript by Gadd et al. entitled "Epigenome and phenome study reveals circulating markers pertinent to brain health" explored associations between protein expression (4,235 plasma proteins) and methylation loci. The authors found 2,895 protein quantitative trait methylation loci (pQTM) after multiple correction. Then the authors evaluated associations between phenotypes in 1,065 subjects with 644 proteins associated with cognitive, brain imaging or APOE status. Finally they integrated the pQTM finding 88 epigenetic associations for protein markers of neurological traits (83 of those new). This is an innovative approach using multiple layers of information for studying brain health. There are some details that may require additional explanation in the text for context to the reader.

Comments:

1. Supplementary table 1: the cognitive scores and brain imaging measures are difficult to interpret. It

will be important to include the units in the tables and some additional information about range and/or distribution (e.g., IQR). Without that differences between the PheWAS and EWAS studies (brain age acceleration and anisotropy) are difficult to understand (specially with those very large SD in several of them).

2. White blood cell proportions (page 5): in addition to the Houseman method, what reference was used (Reinius, Fox, Salas), and what method for the selection of the CpGs used in our cell deconvolution (Jaffe, Koestler). Please add the information to your manuscript.

3. Descriptive information Supplementary table 1 and page 5: It is also important that you summarize the information about your estimated cell proportions for your study, as you are using them to adjust your models, this information is also vital for the context of the manuscript. Similarly additional information about BMI categories, and smoking status is lacking and it is used for your adjusted model. I see also in your methods section that you adjusted for depression, but that information is not summarized in the table either. Please add this data to the manuscript.

4. Effect size (supplementary tables and page 5): The beta coefficients are large, but it is not explained the units in the methylation (beta values, M-values?) and protein (z-scores, log-transformed level, other?). This should be introduced early in the text. What captured my attention is that when compared to Zaghlool et al there are 10 fold differences between both (Supplementary Table 7). This requires some explanation or context for interpretation.

5. Figure 2 and page 6: Could you add some more context to the cis effects, I am confused why 10 Mb? Most of the studies look for closer relationships (1 Mb) around the TSS to locate promoter and enhancer areas (Zaghlool et al use that criterion). In that context trans are limited to TF in different chromosomes and in some cases to changes in large chromosomes (as chromosome 1). Is it possible to elaborate for the reader why that threshold was used and what does that mean for your findings interpretation.

6. Figure 4: The results are well condensed in this circosplot figure pointing to several inflammatory genes. I have only one comment with chromosome 22 as the genes overlap with those of chromosome 21. If there is an alternative to redraw those it will be clearer for your message.

7. I would recommend to add some supplementary plots for the relationships between the two CpGs for NLCR5 and the seven pQTMs. and whether there are particular distributions in those relationships (e.g., genetic components in the association).

8. Limitations: You mention three hypotheses for the non-association between PheWAS and pQTMs. I believe you meant "not reflected by the blood immune cells epigenome" instead of "plasma epigenome".

9. Why the eosinophil proportions were unavailable? Reinius include eosinophils, the newer Salas et al 2021 also incorporates eosinophils. There was any reason not to use it?

10. Methods: DNAm briefly can you report the normalization, p-detection threshold used in your analysis?

11. Code: I briefly looked at the code, and I have questions about your lagged effects and creatinine adjustment. Were those used in your analysis for selection of your sample? I was confused when I found those variables there and not in your manuscript. Could you please clarify?

Reviewer #4 (Remarks to the Author):

The authors perform a systematic study mapping the epigenetic measures to plasma protein levels. The study offers expanded protein measurements compared to an earlier study by Zaghlool et al, Nature Comm, 2020. The novelty here is not in the approach itself, but in the expanded set of pQTM discoveries. The manuscript is overall well-written and I found no major issues in methodology. Whilst the results contain some caveats – this is clearly explained up-front and the authors clearly have gone to some lengths to address this. Some of the caveats are shared with the previous study. Whilst a replication cohort would be nice, I understand the limited availability of such data, and the previous Zaghlool et al study to which the author compares to does provide some indication of replicability.

Overall this work provides a useful resource for the scientific community that expands on existing knowledge in this area assuming summary results will be publicly available.

I have the following comments and questions which I feel needed to be addressed/clarified in a revision:

1. It is possible that the reported numbers may be inflated. I have no issues with the conservative multiple testing threshold. It would be good to know what the correlation structure is like for the CpG measures and how many “effective” independent components drive most the findings here. Also it would be good to know how many proteins are associated in the main results (rather than implied through the GC lambda sentence).
2. The numbers drop drastically once you exclude the 2/3 biggest pleiotropic CpG/proteins, which explains a significant proportion of your 2,854 associations.
3. Of the 151 novel proteins with significant associations, how many are due to the protein not being measured in previous studies.
4. Since effect size are in relative units, I feel sentences relating to effect sizes eg. “There were 2,895 associations, with effect size ranging from -2.64 (SE 0.29) for PRG3 and cg16899419, to 2.62 (SE 0.19) for MDGA1 and cg12415337 (Supplementary Table 5).” are not very informative.
5. Some summary figure/supplementary figure on how many proteins have how many associations/other pleiotropic CpG genes/regions would help interpretability
6. The vast majority of findings seem to be explained by white cell count – I wonder whether other blood cell counts/components may be a confounder in plasma based studies. Do the authors have access to subtype white cell counts, red cell counts, haemoglobin and platelets, that may be adjusted for? Also are cis and trans effects affected differently/similarly with adjustment?
7. I would have though epigenetic effects in theory affect expression rather than post-transcriptional process, at least cis ones? It may be worthwhile to see whether the eQTLs explain some of these associations?
8. I also see ABO there, is this explained by blood group? (assuming there’s access to blood type/genetics to impute the blood type)
9. Despite the lack of replication cohort, the authors attempt to replicate some of the overlap with the existing study. Are any strong associations/relatively pleiotropic associations excluding the ones mentioned, seen in only this study or the other study?
10. What prompts the switch of multiple adjustment methods to FDR for the protein PheWAS rather than stick to one?
11. I believe other studies including Sun et al, Menni et al, Ngo et al also looked at association with age, gender, + others such as BMI/eGFR in addition to Lehallier et al.
12. The proteomic associations with other phenotypes: how much is known and how many are new?

Maybe a forest plot with effect sizes for all/novel proteins rather than an arbitrary selection of scatterplots would be more informative for the space in the figure?

13. "Many of the 644 protein marker associations were independent and did not cross neurological modalities." How is this determined?

14. . Of the 25 common proteins, there were six independent signals, as determined by components with eigenvalues > 1 in principal components analyses (Supplementary Fig.1). How is this justified?

Why use eigenvalue of 1 – doesn't cumulative proportion essentially say the same thing? The first 6 PCs explain less than 70% of the variance. May be an alternative way to cluster may be needed here

15. Are associated genes/proteins systematically enriched for any pathways from your main results?

REVIEWER COMMENTS

General response to reviewers

We thank the reviewers for their comments and suggestions, which have helped us to refine our study design and explore the results further. We have added a number of clarifications and additional analyses based on these reviews that we believe enhance the manuscript. Briefly, the changes include: 1) a tonal adjustment of the scope of our study (including a shift from epigenome-wide to the more DNAm-specific, methylome-wide association study (MWAS) terminology), 2) clarifications of the extent to which pQTM associations can provide biological insights, 3) a full re-run of the MWAS that integrates additional white blood cell estimates, 4) a revision of our multiple testing correction strategy that is informed by principal components analysis of the protein data, 5) further integration of previous research to highlight novel associations, 6) recapitulation of figures to better visualise results with several new supplementary figures, 7) a 10mb vs 1mb thresholding sensitivity analysis for cis associations, 9) an assessment of pleiotropic MWAS associations, 10) FUMA gene set enrichment, FUMA tissue expression and STRING protein interaction analyses to explore candidate pathways implicated by our PheWAS and neurological pQTM results, 11) integration of promoter-capture Hi-C data and CHIP-sequencing marks to support chromatin 3D-genome interactions for the neurological pQTMs and 12) potential genetic component mapping for the neurological pQTMs. We detail these changes in the following point-by-point responses.

General points of note for all reviewers

Open access sharing

Complete summary statistics for all fully-adjusted MWAS will be made available via the MRC-IEU EWAS catalog and this will be included in the manuscript when finalised. As part of this process, the full summary statistics will be hosted on the open-access repository Zenodo. We will also create a YouTube video summarising the findings and detailing how the data can be accessed, which will be hosted on our YouTube account (<https://www.youtube.com/channel/UCxQrFFTItF25YKfJTXuumQ>).

Updated results

When rerunning the study with a more stringent threshold for the protein PheWAS (in response to reviewer 4: comment 10), our total PheWAS associations fell from 644 to 497. Upon reflection, we also decided that protein lag group (time between

sampling and sequencing) and study site (Dundee/Aberdeen) variables should also be included as covariates in the PheWAS. This led to 405 total associations, with 191 proteins associated with one or more of the brain health characteristics. This reduction in protein markers from the PheWAS, combined with further adjustment for eosinophil and monocyte estimates in the MWAS, meant that our total pQTM for neurological protein markers fell from 88 to 35 pQTMs. However, many of the results that inform the core discussion points of the study (SLC7A11, RBL2, SMPD1 and HEXB) remain consistent.

Additional co-authors

We have added Claire Green and Shen Xueyi for their work on the updated imaging variable for white matter hyperintensity volumes that we use in the protein PheWAS. We have added Tamir Chandra and Neil Robertson for their contributions on the CHIP-seq and promoter-focused Hi-C chromatin interaction mapping added in response to points raised by reviewer one.

We have revised the manuscript to reflect the updated analyses – we believe the edits (while substantial) greatly improve the work. This is also true of the Supplementary Figures (that are all new, except one) and Supplementary Methods (that has been added to provide further clarity). We therefore advise (due to the unreadability of the tracked changes version) that the reviewers use a clean copy of the revised manuscript. We provide page and line specificity in our responses for clarity.

Reviewer #1

Comment one

1. Methylation on the CpG sites only represents a fraction of “epigenome”, other types of epigenetic mechanisms, such as 5-hydroxymethylcytosine, histone modifications and chromatin remodeling have all been implicated to coordinatively involve in neurological diseases. On the other hand, increasing evidence suggests critical epigenetic roles of non-CG methylation in neurological diseases. The present study that solely focuses on CpG methylation could be bias and misleading. The term “epigenetics” or “epigenomics” used throughout the paper is inaccurate and confusing.

Response to comment one

We agree that the distinction between DNA methylation at CpG sites and other epigenetic modifications should be clearer. To address this, we have clarified throughout that we perform a methylome-wide association study (MWAS) using only DNAm data at CpG sites. We have also amended our introduction to emphasise that DNAm is one of several possible epigenetic modifications:

Page 3, line 53: *'Epigenetic modifications to the genome record an individual's response to environmental exposures, stochastic biological effects, and genetic influences. Epigenetic changes include histone modifications, non-coding RNA, chromatin remodelling, and DNA methylation (DNAm) at cytosine bases, such as 5-hydroxymethylcytosine. These are implicated in changes to chromatin structure and the regulation of pathways associated with neurological diseases^{11,12}. However, DNAm at cytosine-guanine (CpG) dinucleotides is the most widely profiled blood-based epigenetic modification at large scale.'*

Our discussion has been updated to caveat that when further epigenetic measurements become available at large-scale more widely, alternative modifications beyond DNAm should be assessed:

Page 15, line 349: *'...profiling DNAm signatures alone cannot capture the full role of the epigenome in brain health. Integration of more diverse epigenetic markers will be critical to further resolve these relationships.'*

Comment two part one

1. The authors claimed "As DNAm at CpG sites can affect gene expression in some instances, these relationships may carry information that informs on the regulation of proteins, the biological effector molecules of disease." This is extraordinarily weak as DNA methylation plays differential roles in influencing gene expression at the transcription level, depends on their locations (promoter vs gene bodies, or cis-regulatory elements).

Response to comment two part one

Our original statement was written to avoid mechanistic claims on the role of DNAm at CpGs that associate with protein levels (pQTM). We agree that this is a vague statement when introducing the rationale for studying DNAm. We have therefore updated the text as follows:

Page 3, line 60: *'Modifications to DNAm at CpG sites play differential roles in influencing gene expression at the transcriptional level¹³. Additionally, DNAm accounts for inter-individual variability in circulating protein levels¹⁴⁻¹⁶.*

Comment two part two

2. The post-transcriptional regulation and translation control are additional layers of regulation. Given the DNA methylation in the plasma is not even from the same population of cells or tissues with circulating proteins, the connection investigated in this study does not represent strong biological insights.

Response to comment two part two

We agree that the DNAm is not sourced from the exact same cell types as proteins and we also cannot be sure of the cell type that released proteins into the blood (i.e. liver cells, immune cells, brain cells etc). The SomaScan assay also measures aptamer binding of probes to intracellular, membrane-bound and extracellular protein target epitopes.

A previous EWAS of 1,123 protein levels in blood from Zaghlool *et al*, (*Nat Comms*, 2020) faced the same limitation as our work. This is a known issue that is far-reaching across other EWAS and multi-omics studies. We have added a limitation to our discussion as follows:

Page 14, line 326: *'...the DNAm signatures of proteins we quantify represent widespread differences across blood cells that are related to circulating protein levels and are therefore not derived from the same cell-types as proteins.'*

Nonetheless, there are still important insights from these data types. pQTMs tell us how DNAm from bulk tissue/cells relates to circulating protein levels. pQTMs capture chronic exposure to various environmental and biological states within individuals that associate with protein pathways of interest,

despite us not being certain of the nature of the mechanistic directionality of those relationships.

Previous results from our group suggest that epigenetic scores for circulating proteins (such as SMPD1 – which is also a candidate of interest in this study) constitute signals that are capable of predicting a range of incident diseases (<https://elifesciences.org/articles/71802>). These scores are based on DNAm, indicating that relationships between DNAm and proteins at scale have predictive value for disease risk stratification.

We have updated our manuscript as follows to reflect these points:

Page 3, line 62: *'Recently, through integration of DNAm and protein data, we have shown that epigenetic scores for plasma protein levels – known as 'EpiScores' – associate with brain morphology and cognitive ageing markers¹⁷ and predict the onset of neurological diseases¹⁸. These studies highlight that while datasets that allow for integration of proteomic, epigenetic and phenotypic information are rarely-available, they hold potential to advance risk stratification. Integration may also uncover candidate biological pathways that may underlie brain health.'*

Page 14, line 326: *'Additionally, the DNAm signatures of proteins we quantify represent widespread differences across blood cells that are related to circulating protein levels and are therefore not derived from the same cell-types as proteins. Despite this limitation, previous work supports DNAm scores for proteins as useful markers of brain health, suggesting there is merit in integrating DNAm signatures of protein levels in disease stratification¹⁸.'*

To provide further evidence that the pQTM we identify may represent biological insights (despite the cell-type caveat discussed), we have performed CHIP-seq and promoter-centered Hi-C chromatin mapping, for the 11 neurological pQTMs that involved CpGs within genes that were on the same chromosome as the protein-coding gene. This is discussed further in our response to comment four, however the results indicate that there is modest evidence linking CpG sites with protein-coding gene promoters when separate genes are considered within the 3D genomic neighbourhood. This suggests

that widespread trends in CpG DNAm that correlate with protein levels may indicate functional relationships in some instances. We have also completed additional lookups using UCSC annotation information to further characterise the locations of CpGs in our results. These updates are also discussed in response to comment four below.

While there are methods such as two-sample Mendelian randomisation (MR) that can be used to infer potentially causal relationships between DNAm and gene expression, from our experience this method is impaired by some critical limitations that makes current interpretability challenging:

- 1) The summary statistics for mQTLs/eQTLs mostly report significant QTLs only, so the summary statistics are biased towards those associations
- 2) MR is normally based on Wald ratio in those tests with only single instruments available - because it is a ratio it will likely give a significant association in both directions (i.e. mQTL>eQTL and eQTL>mQTL)
- 3) QTL data are nearly exclusively derived from whole blood, as the reviewer has pointed out

We have therefore chosen to perform an eQTL/mQTL lookup to check for the possible roles of these genetic effects in our pQTM associations. If no mQTL/eQTL signal exists, a DNAm pertaining to a pQTM is more likely to be environmentally-mediated.

Our results section now reads as follows:

Page 13, line 308: *'Given that this study utilised CpGs from the Illumina EPIC array, 15 of the 31 unique CpGs did not have mQTL characterisations in public databases, which primarily comprise results from the earlier 450K array. However, our plots showing pQTM associations suggested that for several CpGs (such as cg11294350 that associated with SMPD1 and HEXB), there may be a partial genetic component influencing DNAm. As mQTLs tend to explain 15-17% of the additive genetic variance of DNAm⁵⁹, it is possible that the signals we isolate in these instances are partially driven by genetic loci, but are also likely driven by unmeasured environmental and biological influences. In the case of SIGLEC5, IL18R1 and CHI3L, mQTLs were identified that were also eQTLs, providing evidence that mQTLs for these CpG sites were possible regulators of protein expression.'*

Comment three

3. The methylated DNA in the plasma could derive from various types of blood cells, secreted vesicles, or any tissues or organs the blood flows. As indicated by the authors, "Blood-based methylation is unlikely to correlate highly with brain methylation in all cases", it is unclear whether the pQTM identified here are specific to neurological diseases showing specific disease manifestations in the brain.

Response to comment three

We thank the reviewer for highlighting this. As we focus on a healthy ageing population as opposed to a case/control study, we agree that generalisation to specific diseases is difficult.

We focus on brain health markers in an ageing population of older adults that are not enriched for neurodegenerative disease. The primary objective of this study is therefore to identify markers that inform on brain health, rather than disease states. However, we anticipate that the findings from our study will help to provide candidate targets that may be investigated in disease states. For example, there is overlap between the proteins and methylation sites we identify here and a range of neurological diseases, which we cover in our discussion. This implies that the markers we have mapped (that are associated with better/poorer neurological health), may represent a pool of possible protein or epigenetic (protein-regulatory) targets that could be considered to improve brain health; this is certainly the case when considering that DNAm at site cg06690548 has been robustly linked to ALS and Parkinson's disease in two studies.

Given that we can't be sure of the source of the methylation or the proteins in this study, we can only make suggestions about possible mechanisms by which these findings could impact on the brain. We have integrated tissue-specific expression mapping for genes corresponding to the proteins implicated in our results. These tissue expression heatmaps are provided for the genes corresponding to protein markers in the PheWAS findings (Supplementary Figure 10) and the genes corresponding to CpGs and proteins involved in neurological pQTMs (Supplementary Figure 13). This provides clarity on whether the proteins are likely to be excreted in brain tissue directly

(and may transition into blood), or whether the markers originate in the periphery (and may impact the brain as a systemic effect), either directly or indirectly influencing brain function via the circulatory system.

We have updated our results for the PheWAS and neurological pQTM as follows:

Page 8, line 176: *'Gene set enrichment analyses on the 191 genes corresponding to the protein markers (Supplementary Fig. 9) supported the link between many of the proteins associated with poorer brain health and the innate immune system, while also implicating extracellular matrix, lysosomal, metabolic and additional inflammatory pathways. Tissue expression profiles of the 191 genes (Supplementary Fig. 10) indicated that many of the markers were expressed non-neurological tissues; however, some proteins were expressed in nervous tissues. Markers such as ASB9 and NCAN were found to be consistently identified across multiple brain imaging traits as markers of poorer and better brain health, respectively (Supplementary Table 16).'*

Page 10, line 228: *'Tissue expression profiles for the 33 genes that were linked to either CpGs or proteins in the 35 neurological pQTMs are summarised in Supplementary Fig. 13. Gene set enrichment for these 33 genes identified enrichment for immune effector pathways in a subset of 11 genes, whereas a cluster of four genes (SMPD1, HEXB, AMY2A and AMY2B) were enriched for amylase and hydrolase activity (Supplementary Fig. 14).'*

Our discussion now reads as follows:

Page 11, line 263: *'Tissue expression analyses suggested that a large proportion of the 191 protein markers were not expressed in the brain; this supports work suggesting that sustained peripheral inflammation influences general brain health^{31,32} and accelerates cognitive decline^{8,33–35}. However, a subset of proteins were expressed in the central nervous system. Given that leakage at the blood-brain-barrier interface has been hallmarked as a part of healthy brain ageing^{36,37}, there is a possibility that brain-derived proteins may enter the bloodstream as biomarkers. SLIT and NTRK Like Family Member 1 (SLITRK1), Neurocan (NCAN) and IgLON family member 5 (IGLON5) were examples of proteins expressed in brain for which higher levels associated with either larger grey matter volume, larger whole brain volume, or higher general fractional anisotropy. SLITRK1*

*localises at excitatory synapses and regulates synapse formation in hippocampal neurons*³⁸. Neurocan (NCAN) is a component of neuronal extracellular matrix and is linked to neurite growth³⁹. IGLON5 has been implicated in maintenance of blood-brain-barrier integrity and an anti-IGLON5 antibody disease involves the deterioration of cognitive health⁴⁰. Taken together, the protein markers identified in the PheWAS may, therefore, reflect pathways that could be targeted to improve brain health.'

Page 13, line 296: *'Many of the genes corresponding to CpGs and proteins involved in the 35 pQTM were enriched for immune effector processes and were not expressed in brain. However, some markers did show evidence for brain-specific expression, such as acid sphingomyelinase (SMPD1) and Hexosaminidase Subunit Beta (HEXB).'*

Regarding DNAm consistencies between blood and brain as the tissue of interest, while looking at blood-brain correlations (using a publicly available DNAm comparison tool - <https://epigenetics.essex.ac.uk/bloodbrain/>) is useful, this tool does not integrate the most recent Illumina EPIC array (with 800,000+ probes), nor does it interrogate all brain regions. It is also worth noting that a lack of correlation between DNAm at CpGs between blood and brain is not necessarily equivalent to the blood findings being irrelevant to the brain. There are signatures that will be specific to blood that are still of predictive value for brain health; for example, our recent work has demonstrated that a blood-based signature of DNAm for SMPD1 is able to predict onset of Alzheimer's dementia (Gadd *et al*, 2022). We show that blood patterns of DNAm are able to delineate risk, implying that there is value in understanding DNAm markers in the blood that are specific to the biological state of individuals with certain protein levels in circulation.

We have integrated promoter-capture Hi-C data from the brain (hippocampus) and CHIP-sequencing data (from both blood and brain) in our analyses. The mapping has been conducted for the 11 neurological *cis* pQTMs with CpGs and proteins that lie within genes located on the same chromosome. We show that there are common CHIP-sequencing marks across blood and brain for the genes of interest. We also find evidence that brain hippocampal tissue has chromatin interactions for genes implicated in our pQTM results. This supports the possibility that the pQTMs we identify may have relevance to the brain. These analyses are detailed in response to

comment four. We have added the following discussion point regarding blood and brain differences to the manuscript:

Page 15, line 346: *'...differences in blood and brain DNAm and pQTLs are emerging; these indicate that blood-based markers may not fully align to biology of brain degeneration^{65,66}. However, our ChIP-seq analysis of chromatin regulation suggested that some regulatory states may persist between blood and brain.'*

Comment four

4. For the 255 trans CpG that more than 10Mb from the transcription start site of the protein gene (Fig. 2), the direct transcription regulation is unlikely. Do those CpG sites locate in known cis-regulatory elements to form 3D genomic interaction with these protein targets?

Response to comment four

The reviewer makes an excellent point here. To address this, we have carried out a series of CHIP-sequencing and promoter-capture Hi-C analyses. We focus on the 35 pQTLs, where the protein is associated with both DNAm and a brain health outcome.

We have split our pQTLs into the following subgroups when considering possible functional pathways:

- Cis associations with CpGs proximal to the protein-coding gene
- Cis associations on the same chromosome, but with different CpG and protein genes
- Trans associations on different chromosomes, with different CpG and protein genes

Due to chromosomal segregation, inter-chromosomal contacts tend to be more stochastic and less persistent, ergo, we have chosen to interrogate intra-chromosomal *cis*- associations (involving different CpG and protein genes) with ChIP-seq and promoter-capture Hi-C mapping to assess whether there might be some basis for shared regulation and chromatin interaction in these

loci. The mapping for each of the seven proteins implicated in 11 pQTM is presented in Supplementary Figures 15-21.

Our methods section now reads as follows:

Page 22, line 518: *'Although inter-chromosomal chromatin interactions are unlikely to be stable and persistent, seven proteins with cis pQTMs involving CpGs located intra-chromosomally to the proximal protein-coding gene were considered for ChIP-seq and promoter-capture Hi-C mapping to interrogate local chromatin interactions and states that might form the basis for co-regulation of these loci. ChIP-seq data from peripheral blood mononuclear cells (PBMCs) and brain hippocampus were selected from the ENCODE project ⁸⁶, with accession identifiers available in Supplementary Table 21. Processed promoter-capture Hi-C data for brain hippocampal tissue was integrated from Jung et al, ⁶⁰ and is available at NCBI Geo with accession GSE86189. Data concerning both promoter-promoter interactions and promoter-other interactions were concatenated and all regions subsequently visualised on the WashU epigenome browser ⁸⁷.'*

Our results section now reads as follows:

Page 10, line 237: *'Promoter-capture Hi-C and ChIP-sequencing integration was used to assess the interactions and chromatin states of our pQTMs and associated CpG loci. This analysis focused on 11 of the 20 cis pQTMs that involved CpGs on the same chromosome as the protein-coding gene, but were located in a different gene. Mapping information is presented for the seven proteins involved in these pQTMs in Supplementary Figs. 15-21. In all instances, we found evidence of spatial co-localisation of these genes using promoter-capture Hi-C data from brain hippocampal tissue. We attempted to contextualise these sites further with ChIP-seq (ENCODE project) analyses of active chromatin marks H3K27ac and H3K4me1 and repressive chromatin H3K4me3 and H3K27me3 in both peripheral blood mononuclear cells (PBMCs) and brain hippocampus. ChIP-seq data suggested that in many instances there were shared regulatory regions that existed across both blood and hippocampal samples that were hubs for local promoter interactions. For example, promoter loops were found linking the S100Z and CRHBP genes, with a signature of activating (H3Kme1 and H3K27ac) and silencing (H3k27me and H3K4me3) marks (normally considered bivalent chromatin) that may form the basis for shared regulation of this gene locus.'*

Our discussion reads as follows:

Page 14, line 317: *'Integration of promoter-capture Hi-C chromatin interaction and ChIP-seq databases ⁶⁰ provided evidence for long-range interaction relationships for cis pQTM with CpGs in different gene regions that are proximal to the protein-coding gene of interest. This suggests that in such instances, the pQTM may reflect regulatory relationships in the 3-dimensional genomic neighbourhood. The pQTM therefore direct us towards pathways that can be tested in experimental constructs. Positional information suggested that many CpGs involved in neurological pQTM lay within 1500 bp of the TSS of the respective protein-coding gene. While positional information of CpGs is thought to infer whether DNAm is likely to play a role in the expression regulation of nearby genes, this is still somewhat disputed. Some studies suggest that transcription factors regulate DNAm ⁶¹ and differential methylation at gene body locations predicts dosage of functional genes ⁶². Additionally, the DNAm signatures of proteins we quantify represent widespread differences across blood cells that are related to circulating protein levels and are therefore not derived from the same cell-types as proteins. Despite this limitation, previous work supports DNAm scores for proteins as useful markers of brain health, suggesting there is merit in integrating DNAm signatures of protein levels in disease stratification ¹⁸.*

To add greater positional context to the CpGs implicated in our wider pQTM results, we examined positional information through UCSC database annotations. We have integrated all lookup results into the revised Supplementary Table 6 and Supplementary Table 20.

We present UCSC lookups of the CpGs implicated in pQTM as follows:

Page 5, line 115: *'Characterising the genomic location of the findings, 46% of cis and 29% of trans pQTM in the fully-adjusted MWAS involved CpGs positioned in either a CpG Island, Shore or Shelf (Supplementary Table 6).'*

Page 10, line 233: *'Of the 35 pQTM, seven had CpGs that were located in either a CpG Shore or Shelf position and there were 13 that were located either 1500 bp or 200 bp from the TSS of the protein-coding gene (Supplementary Table 20). Fifteen pQTM involved CpGs that were located in the gene body and 7 were located in either the first exon or UTR regions (Supplementary Table 20).'*

These approaches guide us on positional context of CpGs within islands/shores/shelves, their relation to the transcriptional start site of the gene encoding the protein and the likelihood of the region being a site of transcription factor (TF) binding. There is still debate in the field as to whether the assumption that DNAm located at promoter regions always regulates TF binding is representative – as some evidence also suggests that TFs may regulate DNAm. Additionally, we also note that CpGs in the body of genes have been shown to influence gene dosage.

We have therefore added the following section to our discussion:

Page 14, line 321: *'Positional information suggested that many CpGs involved in neurological pQTM lay within 1500 bp of the TSS of the respective protein-coding gene. While positional information of CpGs is thought to infer whether DNAm is likely to play a role in the expression regulation of nearby genes, this is still somewhat disputed. Some studies suggest that transcription factors regulate DNAm⁶¹ and differential methylation at gene body locations predicts dosage of functional genes⁶².*

We recognise however that the cell-type limitation discussed is a major limitation to any functional mapping of likely relationships between DNAm and protein levels. For this reason, we state the following limitation in our discussion:

Page 14, line 326: *'Additionally, the DNAm signatures of proteins we quantify represent widespread differences across blood cells that are related to circulating protein levels and are therefore not derived from the same cell-types as proteins. Despite this limitation, previous work supports DNAm scores for proteins as useful markers of brain health, suggesting there is merit in integrating DNAm signatures of protein levels in disease stratification¹⁸.*

Reviewer #2

Comment one

1. Supplementary table 1: the cognitive scores and brain imaging measures are difficult to interpret. It will be important to include the units in the tables and some

additional information about range and/or distribution (e.g., IQR). Without that differences between the PheWAS and EWAS studies (brain age acceleration and anisotropy) are difficult to understand (especially with those very large SD in several of them).

Response to comment one

We have updated the now revised Supplementary Table 2 to show the IQR for each of the cognitive and brain imaging measures used. We also agree with the reviewer that we could do more to describe the range and distributions for the phenotypes studied. We therefore plot the distributions of the cognitive and brain imaging measures used in our study and include these as the revised Supplementary Figures 4-5. We have also provided a more detailed overview of the imaging and cognitive phenotype measurements in the revised Supplementary Methods. We hope that in combination, these updates sufficiently describe the data for interpretation.

Regarding the query on units, the units for brain imaging variables have been added where appropriate to do so. An overview is given for each measure:

- Predicted Brain Age – this variable is the brain age estimate generated (in years)
- Brain Age Acceleration – this variable describes differences between predicted brain age and chronological age and is therefore in years
- Global Grey Matter Volume, Whole Brain Volume and Intracranial Volume - these are derived with the FreeSurfer toolkit and are therefore in cubic millimeters (mm³)
- White Matter Hyperintensity Volume – this variable is measured in milliliters (one unit is 1000 mm³)
- Global Fractional Anisotropy / Global Mean Diffusivity – these measures are derived from standardized FA and MD measures for individual white matter tracts with PCA and are thus 'unitless'

With regard to the cognitive tests we have updated our methods and Supplementary Methods as follows (key changes highlighted in yellow) to include the maximum scores possible for each test, in addition to what each test measures to create scores:

Page 18, line 410: 'Full details for the specific scores has been detailed previously⁶⁸ and further details can be found in Supplementary Methods. Briefly, these included the Wechsler Logical Memory Test (maximum possible score of 50), the Wechsler Digit Symbol Substitution Test (maximum possible score of 133), the verbal fluency test (based on the Controlled Oral Word Association task), the Mill Hill Vocabulary test (maximum possible score of 44) and the Matrix Reasoning test (maximum possible score of 15).'

Supplementary Methods: 'The sum of immediate and delayed recall of one oral story from the Wechsler Logical Memory Test was taken as the logical memory phenotype (maximum score of 25 for each recall test with a combined maximum score of 50)⁶. Details about the stories that were remembered correctly were recorded as points contributing to the scores. The Wechsler Digit Symbol Substitution Test, which requires individuals to recode digits to symbols and represents a count of correct pairs within a timeframe of 120 seconds was used to measure processing speed phenotype⁷. The verbal reasoning phenotype measures verbal comprehension and phonemic fluency and was based on the Controlled Oral Word Association task⁸ with letters C, F and L. The number of words named starting with the given letters in a one minute period were recorded as the score. The Mill Hill Vocabulary test was used as a measure of acquired verbal intelligence, and is an estimate of 'crystallised intelligence' and peak cognitive ability. This records the number of times participants successfully explain the meaning of words select a synonym, using junior and senior synonyms⁹. The Matrix Reasoning test, a paper adaptation of the computerised version from the COGNITO psychometric examination¹⁰ was used to measure perceptual organisation and visuospatial logic. The Matrix Reasoning test measures non-verbal, abstract reasoning and records the number of correct answers recognising the missing element in a pattern that is presented as a matrix.'

The gf and g measures are derived from principal components of the scores in combinations, and are standardised scores with mean 0 and variance of 1.

Comment two

2. White blood cell proportions (page 5): in addition to the Houseman method, what reference was used (Reinius, Fox, Salas), and what method for the selection of the CpGs used in your cell deconvolution (Jaffe, Koestler). Please add the information to your manuscript.

Response to comment two

We have updated our 'Phenotypes in STRADL' methods section to clarify this as follows:

Page 17, line 397: *'The meffil⁶⁸ implementation of the Houseman method was used to calculate estimated white blood cell (WBC) proportions for Sets 1 and 2. Blood reference panels were sourced from Reinus et al⁶⁹. The 'blood gse35069 complete' panel was used to imputed measures for Monocytes, Natural Killer cells, Bcells, Granulocytes, CD4⁺T cells and CD8⁺T cells. Eosinophil and Neutrophil estimates were also sourced through the 'blood gse35069' panel.'*

Comment three

3. Descriptive information Supplementary table 1 and page 5: It is also important that you summarize the information about your estimated cell proportions for your study, as you are using them to adjust your models, this information is also vital for the context of the manuscript. Similarly additional information about BMI categories, and smoking status is lacking and it is used for your adjusted model. I see also in your methods section that you adjusted for depression, but that information is not summarized in the table either. Please add this data to the manuscript.

Response to comment three

Thank you for raising this. We have added summary information for depression, estimated cell proportions and BMI as additional variables to the now revised Supplementary Table 2. Given we adjusted our epigenetic-association study using the DNAm score for smoking (EpiSmokEr), we have added the mean, sd and IQR for this score for this group to the table. We have also ensured these variables are summarised appropriately in the 'Phenotypes in Generation Scotland' section of the methods in our manuscript on page 17.

Comment four

4. Effect size (supplementary tables and page 5): The beta coefficients are large, but it is not explained the units in the methylation (beta values, M-values?) and protein (z-scores, log-transformed level, other?). This should be introduced early in the text.

What captured my attention is that when compared to Zaghlool et al there are 10 fold differences between both (Supplementary Table 7). This requires some explanation or context for interpretation.

Response to comment four

In our epigenome-wide association study we used rank-based inverse normalised protein levels and M-values for our DNAm dataset. Zaghlool et al used log transformed protein levels with M-values. However, P-values and direction of effects are consistent across both studies – see Supplementary Table 11.

To aid in the comparison given in Supplementary Table 11, we have converted the effect estimates from both studies to z-score format and plotted the replication in Supplementary Figure 3. We hope that this further visualisation is of value.

Comment five

5. Figure 2 and page 6: Could you add some more context to the *cis* effects, I am confused why 10 Mb? Most of the studies look for closer relationships (1 Mb) around the TSS to locate promoter and enhancer areas (Zaghlool et al use that criterion). In that context *trans* are limited to TF in different chromosomes and in some cases to changes in large chromosomes (as chromosome 1). Is it possible to elaborate for the reader why that threshold was used and what does that mean for your findings interpretation.

Response to comment five

To understand the impact that this decision has on our results, we have performed a sensitivity assessment that restricted the distance to within 1Mb of the TSS for the genes corresponding to the fully-adjusted pQTM associations. We restricted *cis* associations to CpGs on the same chromosome as the protein, as per our main analyses. This analysis suggests that the WBC-adjusted and fully-adjusted *cis* classifications are largely unaffected by differences in the distance threshold used.

In this assessment, we found that:

- Of the 451 associations from the fully-adjusted MWAS that were classified as cis at the 10Mb threshold, 409 cis associations fell within the 1Mb distance from the TSS, while 42 were within the 10Mb-1Mb region of the TSS
- Of the 453 associations from the WBC-adjusted MWAS that were classified as cis at the 10Mb threshold, 413 cis associations fell within the 1Mb distance from the TSS, while 40 were within the 10Mb-1Mb region of the TSS
- Of the 2,107 associations from the basic MWAS that were classified as cis at the 10Mb threshold, 752 cis associations fell within the 1Mb distance from the TSS, while 1355 were within the 10Mb-1Mb region of the TSS

We have added this sensitivity analysis to our results section:

Page 5, line 111: *'In a sensitivity analysis, restriction of the threshold for cis pQTM from 10Mb to 1Mb from the transcription start site of the gene encoding the protein yielded 409 cis pQTMs (a drop of 42 pQTMs) in the fully-adjusted MWAS.'*

We have clarified our cis/trans definitions in the methods as follows:

Page 22, line 500: *'pQTMs were classified as cis if the CpG was on the same chromosome as the protein-coding gene and fell within 10Mb of the transcriptional start site (TSS) of the protein gene. pQTMs involving a CpG located on a different chromosome to the protein-coding gene, or >10Mb from the TSS of the protein gene were classed as trans.'*

We have also ensured that this is made clear in the Figure 2 part b legend:

Fig 2 legend: *"The 434 cis pQTMs (purple) lay on the same chromosome and \leq 10Mb from the transcriptional start site (TSS) of the protein gene, whereas the 391 trans pQTMs (green) lay > 10Mb from the TSS of the protein gene or on a different chromosome."*

We agree that there is variable use of the 1Mb and 10Mb threshold in the literature. The 10mb threshold has been used in two previous studies by our

group, which quantified genetic and epigenetic associations with protein levels:

- <https://genomemedicine.biomedcentral.com/articles/10.1186/s13073-020-00754-1>
- <https://www.nature.com/articles/s41467-019-11177-x>

While there are also epigenetic and genetic studies (such as Sun et al) that restrict cis associations to within 1Mb of the transcription start site (TSS) of the gene encoding the protein, there are arguably cis associations that may fall slightly beyond such distances. Our choice to use a more lenient 10Mb threshold is based on this rationale.

Comment six

6. Figure 4: The results are well condensed in this circosplot figure pointing to several inflammatory genes. I have only one comment with chromosome 22 as the genes overlap with those of chromosome 21. If there is an alternative to redraw those it will be clearer for your message.

Response to comment six

We have updated this figure based on the revised findings. Due to the number of neurological pQTM's dropping from 88 to 35 as a result of immune cell adjustments in the MWAS and more stringent thresholding adjustment for multiple testing in the PheWAS, the text is now much larger and should help with interpretability for readers.

Comment seven

7. I would recommend to add some supplementary plots for the relationships between the two CpGs for NLCR5 and the seven pQTM's. and whether there are particular distributions in those relationships (e.g., genetic components in the association).

Response to comment seven

We have used the DNAm and protein datasets that fed into the MWAS to plot the relationships for the revised set of pQTM's for all 35 associations in the neurological subset, in the interests of thoroughness and visualisation. These

plots are included in Supplementary Figure 12. Several of the distributions indicated the presence of trimodal distributions in CpG DNAm, implying they have an underlying genetic component. We performed a lookup of mQTLs/eQTLs in public databases, however due to these relying on data from the 450K Illumina array, this search was limited to partial coverage of the relevant CpGs (15/31 CpGs were unique to the EPIC array).

The results section has been updated as follows:

Page 10, line 221: *'A lookup that integrated information from the GoDMC and eQTLGen databases assessed whether pQTLs were partially driven by an underlying genetic component. This identified methylation quantitative trait loci (mQTLs) for CpGs that were associated with CHI3L1, IL18R1 and SIGLEC5 and were also expression quantitative trait loci (eQTLs) for the respective protein levels (Supplementary Table 20). Further visual inspection of the distributions for the 35 pQTLs indicated that trimodal distributions – suggestive of unaccounted SNP effects – were present for CpGs involved in seven of the pQTLs (Supplementary Fig. 12).'*

We have also discussed the interpretation of these genetic components as follows, referencing the plots and mQTL/eQTL analyses:

Page 13, line 308: *'Given that this study utilised CpGs from the Illumina EPIC array, 15 of the 31 unique CpGs did not have mQTL characterisations in public databases, which primarily comprise results from the earlier 450K array. However, our plots showing pQTL associations suggested that for several CpGs (such as cg11294350 that associated with SMPD1 and HEXB), there may be a partial genetic component influencing DNAm. As mQTLs tend to explain 15-17% of the additive genetic variance of DNAm⁵⁹, it is possible that the signals we isolate in these instances are partially driven by genetic loci, but are also likely driven by unmeasured environmental and biological influences. In the case of SIGLEC5, IL18R1 and CHI3L, mQTLs were identified that were also eQTLs, providing evidence that mQTLs for these CpG sites were possible regulators of protein expression.'*

Comment eight

8. Limitations: You mention three hypotheses for the non-association between PheWAS and pQTM. I believe you meant "not reflected by the blood immune cells epigenome" instead of "plasma epigenome".

Response to comment eight

Thank you for pointing this out. We have updated the sentence to the following:

Page 15, line 340: *'1) the presence of pathways relating to neurological disease that are not reflected by blood immune cell DNAm...'*

Comment nine

9. Why the eosinophil proportions were unavailable? Reinius include eosinophils, the newer Salas et al 2021 also incorporates eosinophils. There was any reason not to use it?

Response to comment nine

Thank you for raising this point. At the time of rerunning our analyses, we generated the eosinophil estimates using Reinius et al 2012 as the reference panel. We also were able to source Neutrophil estimates from this panel. When correlating measurements for all estimates white blood cell proportions available (Supplementary Figure 22), we observed that the neutrophil estimates were highly correlated (>95%) with granulocyte estimates. For this reason, we chose to include Monocytes, Bcells, CD4T cells, CD8T cells, Granulocytes, Natural Killer cells and Eosinophils in our updated MWAS in WBC-adjusted and fully-adjusted models. Previously we had used Bcells, CD4T cells, CD8T cells, Granulocytes and Natural Killer cells. Our new results therefore have further adjustment for Monocyte and Eosinophil estimates and we see a reduction in pQTMs, suggesting that our previous associations for MX1 were likely to have been driven by the cell type effects.

Page 21, line 483: *'A second model further adjusted for estimated white blood cell proportions (Monocytes, CD4⁺T cells, CD8⁺T cells, BCells, Natural Killer cells, Granulocytes and Eosinophils). While Neutrophil estimates were available, they*

were excluded due to high correlation (>95%) with Granulocyte proportions (Supplementary Fig. 22).'

We appreciate that a more recent study is now available that facilitates eosinophil estimates (Salas et al, 2021), however, given that was in the form of a preprint when we ran our MWAS analyses we chose not to use it on this occasion. However, we have noted it as a reference in our limitations/future applications as follows:

Our updated text reads as follows:

Page 14, line 334: *'While comprehensive adjustment for estimated immune cells was performed and the remainder of CpGs involved in pQTM did not show high correlations (Supplementary Fig. 2), concurrently measured blood components such as haemoglobin, red blood cells and platelets were not available. Future studies should seek to resolve signals with more detailed blood-cell phenotyping and immune cell estimates⁶³.*

Comment ten

10. Methods: DNAm briefly can you report the normalization, p-detection threshold used in your analysis?

Response to comment ten

We have added this information and also provide more extensive details regarding the DNAm preparation as a subsection of the newly included Supplementary Methods file. Our updated manuscript now reads as follows:

Page 17, line 383: *'Measurements of blood DNAm in the STRADL subset of GS subset were processed in two sets on the Illumina EPIC array using the same methodology as those collected in the wider Generation Scotland cohort. Quality control details have been reported previously⁷⁰⁻⁷² and further details are provided in Supplementary Methods. Briefly, samples were removed if there was a mismatch between DNAm-predicted and genotype-based sex and all non-specific CpG and SNP probes (with allele frequency > 5%) were removed from the methylation file. Probes which had a beadcount of less than 3 in more than 5% of samples and/or probes in which >1% of samples had a detection $P > 0.01$*

were excluded. After quality control, 793,706 and 773,860 CpG were available in sets 1 and 2, respectively.'

Comment eleven

11. Code: I briefly looked at the code, and I have questions about your lagged effects and creatinine adjustment. Were those used in your analysis for selection of your sample? I was confused when I found those variables there and not in your manuscript. Could you please clarify?

Response to comment eleven

Thank you for raising this important point. We apologise that our code was not presented clearly enough for full interpretation. In a previous iteration of the study we had initially integrated eGFR as a covariate and adjusted the protein levels for this measure. However, we realised that the eGFR was sampled at Generation Scotland baseline. Given the STRADL sample was taken ~5 years after baseline, we decided that this variable should not be used in analyses. The protein preparation script that you viewed had the remnants of the eGFR preparation step, however we did not use the eGFR variable in the regression onto protein levels for the MWAS that was detailed in the script.

In response to this issue, we have checked all code presented on GitLab, to ensure that it is true to the final analyses we present in the paper. We will be producing a YouTube video that informs people on both the open access data and code that is available.

Reviewer #4

Comment one

1. It is possible that the reported numbers may be inflated. I have no issues with the conservative multiple testing threshold. It would be good to know what the correlation structure is like for the CpG measures and how many "effective" independent components drive most the findings here. Also it would be good to know how many proteins are associated in the main results (rather than implied through the GC lambda sentence).

Response to comment one

We agree that PAPPa and PRG3 had a significant proportion of the pQTM associations and were likely to be inflated. This was aligned to Zaghlool *et al* (*Nat Comms*, 2020), which found that PAPPa was also the largest component of associations in their study (72 of a possible 98 pQTMs). PRG3 was not measured by Zaghlool *et al*, which is why we do not see high inflation for this protein in their results.

We have made the distinction between PAPPa/PRG3 and the rest of the protein associations clearer throughout our results section (see Fig. 2b). We have also add a limitation to our discussion to discuss the potential for inflation as follows:

Page 14, line 333: *'...we observed a substantial inflation for PAPPa and PRG3 proteins.'*

Regarding the second point raised on 'effective independent components', we now present the correlation structure for CpG measures with a PCA analysis to show how many independent signals drive the majority of findings in Supplementary Figure 2. These indicated a high level of intercorrelations for the inflated proteins (PRG3 and PAPPa). We have updated the text as follows:

Page 6, line 123: *'Principal components analyses indicated high correlations between CpGs associated with the pleiotropic proteins PAPPa and PRG3, whereas the CpGs involved in the remaining 825 pQTMs were largely uncorrelated (Supplementary Fig. 2).'*

Finally, we have clarified the number of proteins implicated in our results, separate to the sentence that refers to lambda values. This text now reads as follows:

Page 5, line 108: *'There were 191 unique proteins with associations in the fully-adjusted models, corresponding to 195 SOMAmer measurements (two SOMAmers were present for CLEC11A, GOLM1, ICAM5 and LRP11). Genomic inflation statistics for these 195 SOMAmer measurements (fully-adjusted MWAS) are presented in Supplementary Table 7.'*

Comment two

2. The numbers drop drastically once you exclude the 2/3 biggest pleiotropic CpG/proteins, which explains a significant proportion of your 2,854 associations.

Response to comment two

As mentioned in the response to the above comment, we have made this distinction clearer throughout the manuscript. Our revised visualisations (Fig. 2b and Fig3) also make the distinction clearer.

We have also added a limitation on the potential for inflation as follows:

Page 14, line 334: *'...Second, we observed a substantial inflation for PAPPA and PRG3 proteins. While comprehensive adjustment for estimated immune cells was performed and the remainder of CpGs involved in pQTM did not show high correlations (Supplementary Fig. 2), concurrently measured blood components such as haemoglobin, red blood cells and platelets were not available. Future studies should seek to resolve signals with more detailed blood-cell phenotyping and immune cell estimates⁶³.*

Comment three

3. Of the 151 novel proteins with significant associations, how many are due to the protein not being measured in previous studies.

Response to comment three

We have updated our manuscript as follows to clarify this:

Page 6, line 137: *'When accounting for 26 pQTMs that were previously reported by Zaghlool et al and 10 pQTMs that were previously reported by Hillary et al^{14,19}, 2,892 of the 2,928 fully-adjusted pQTMs were novel. Of these 2,892 novel pQTMs, 1,109 involved the levels of 41 proteins that were measured by Zaghlool et al (973 pQTMs for PAPPA and 136 additional pQTMs for the levels of 40 proteins), whereas 1,783 pQTMs involved the levels of proteins that*

were previously unmeasured (1,116 pQTM for PRG3 and 667 further pQTMs for 148 proteins).'

Comment four

4. Since effect size are in relative units, I feel sentences relating to effect sizes eg. "There were 2,895 associations, with effect size ranging from -2.64 (SE 0.29) for PRG3 and cg16899419, to 2.62 (SE 0.19) for MDGA1 and cg12415337 (Supplementary Table 5)." are not very informative.

Response to comment four

We agree that this sentence is uninformative and have removed it. This has been replaced with further clarification on the strongest/most pleiotropic associations as requested in comments five and nine.

Comment five

5. Some summary figure/supplementary figure on how many proteins have how many associations/other pleiotropic CpG genes/regions would help interpretability

Response to comment five

We have added a visualisation that forms the revised Fig. 3 to highlight the most pleiotropic proteins/CpGs involved in the fully-adjusted MWAS pQTMs. This figure plots the associations with protein chromosomal locations on the x-axis and CpG chromosomal locations on the y-axis. In each case, we highlight the proteins or CpGs that had the most pleiotropic signatures in the results. We also supplement this by presenting counts for associations in Supplementary Tables 9-10. Part c of Figure 2 also presents a flow diagram showing the number of associations that were found in the fully-adjusted MWAS, with the highly pleiotropic associations for PAPP A and PRG3 separated out from the remaining 825 pQTMs for 189 protein levels. Though it is challenging to present such a large amount of information in a diagram such as this, we hope that these amends make the findings clearer.

Comment six

6. The vast majority of findings seem to be explained by white cell count – I wonder whether other blood cell counts/components may be a confounder in plasma based studies. Do the authors have access to subtype white cell counts, red cell counts, haemoglobin and platelets, that may be adjusted for? Also are cis and trans effects affected differently/similarly with adjustment?

Response to comment six

Regarding the first point, we unfortunately do not have access to red cell counts, haemoglobin and platelet measurements at the same time as DNAm in the Generation Scotland cohort. Though we have adjusted for white blood cell estimates, it is possible that there may be other blood-based cell composition factors that are unaccounted for by this study that may influence pQTM associations. For this reason, we have added a caveat as follows:

Page 14, line 334: *'...Second, we observed a substantial inflation for PAPPA and PRG3 proteins. While comprehensive adjustment for estimated immune cells was performed and the remainder of CpGs involved in pQTMs did not show high correlations (Supplementary Fig. 2), concurrently measured blood components such as haemoglobin, red blood cells and platelets were not available. Future studies should seek to resolve signals with more detailed blood-cell phenotyping and immune cell estimates⁶³.*

With regards to the second point raised, we agree that the associations that are accounted for by adjustment using white blood cell estimates are interesting. As the MWAS was previously run with WBC and pQTL adjustment in the same model, we have now included pQTL adjustments in our basic model. This has allowed us to report the associations that are attenuated due to WBC adjustment specifically. We have updated the manuscript as follows with cis/trans breakdowns for each stage of the MWAS with cis/trans information also now included in the Supplementary Tables for each MWAS iteration:

Page 5, line 99: *'In our basic model adjusting for age, sex and available genetic pQTL effects from Sun et al²⁰ 238,245 pQTMs (2,107 cis and 236,138 trans, representing 0.005% of tested associations) had $P < 4.5 \times 10^{-10}$ (Supplementary*

Table 4). In our second model that further adjusted for Houseman-estimated white blood cell proportions²¹, there were 3,213 associations (453 cis and 2,760 trans) that had $P < 4.5 \times 10^{-10}$ (Supplementary Table 5). Smoking status and BMI are known to have well-characterised DNAm signatures^{22,23}; fully-adjusted models were therefore further adjusted for these factors. There were 2,928 associations (451 cis and 2,477 trans) in the fully-adjusted models (Supplementary Table 6). 2,847 pQTM associations were significant in all models. Figure 2 summarises these findings.'

Comment seven

7. I would have thought epigenetic effects in theory affect expression rather than post-transcriptional process, at least cis ones? It may be worthwhile to see whether the eQTLs explain some of these associations?

Response to comment seven

Given that our DNAm has been measured on the Illumina EPIC array, we do not have access to any public databases that would allow us to search for mQTLs across all CpGs included in the present study. We have performed a lookup of both mQTLs (from the GoDMC database) and eQTLs (from the eQTLGen database) with partial coverage. This identified three proteins that may be driven by an mQTL that is also an eQTL. We include this in Supplementary Table 20 (for the neurological subset of proteins with pQTM signals). We also plotted the pQTM associations for the 35 pQTMs associated with brain health protein markers (Supplementary Figure 12). This revealed seven associations that had CpGs with possible mQTL effects via trimodal distributions of the CpG DNAm. Of the 31 unique CpGs, 15 are unique to the EPIC array and therefore not included in the GoDMC database.

We detail this in our results as follows:

Page 10, line 221: *A lookup that integrated information from the GoDMC and eQTLGen databases assessed whether pQTMs were partially driven by an underlying genetic component. This identified methylation quantitative trait loci (mQTLs) for CpGs that were associated with CHI3L1, IL18R1 and SIGLEC5 and were also expression quantitative trait loci (eQTLs) for the respective protein levels (Supplementary Table 20). Further visual inspection of the distributions for*

the 35 pQTM indicated that trimodal distributions – suggestive of unaccounted SNP effects – were present for CpGs involved in seven of the pQTMs (Supplementary Fig. 12).'

We also discuss this as follows:

Page 13, line 308: *'Given that this study utilised CpGs from the Illumina EPIC array, 15 of the 31 unique CpGs did not have mQTL characterisations in public databases, which primarily comprise results from the earlier 450K array. However, our plots showing pQTM associations suggested that for several CpGs (such as cg11294350 that associated with SMPD1 and HEXB), there may be a partial genetic component influencing DNAm. As mQTLs tend to explain 15-17% of the additive genetic variance of DNAm⁵⁹, it is possible that the signals we isolate in these instances are partially driven by genetic loci, but are also likely driven by unmeasured environmental and biological influences. In the case of SIGLEC5, IL18R1 and CHI3L, mQTLs were identified that were also eQTLs, providing evidence that mQTLs for these CpG sites were possible regulators of protein expression.'*

Comment eight

8. I also see ABO there, is this explained by blood group? (assuming there's access to blood type/genetics to impute the blood type)

Response to comment eight

We have sourced blood type information from the Generation Scotland study. However, since a more stringent thresholding approach was applied to the protein PheWAS to adjust for multiple testing, ABO is now no longer implicated as a protein marker for neurological traits. This additional analysis for the neurological pQTM pertaining to ABO is now no longer required.

Comment nine

9. Despite the lack of replication cohort, the authors attempt to replicate some of the overlap with the existing study. Are any strong associations/relatively pleiotropic associations excluding the ones mentioned, seen in only this study or the other study?

Response to comment nine

As the only other large-scale MWAS on circulating proteins was conducted by Zaghlool et al, we compared strong/pleiotropic associations across both studies. Our updated replication assessment (Supplementary Figure 3 and Supplementary Table 11) shows that we replicate all pQTM comparable with Zaghlool et al at the nominal threshold of $P < 0.05$ and we replicate many of the strongest associations at $P < 4.5 \times 10^{-10}$.

Regarding pleiotropy across both studies, we replicate the highly pleiotropic associations found by Zaghlool et al for PAPP A protein levels, and for CpGs in the NLRC5 gene region. Our results also found that PRG3 was a highly pleiotropic protein – this protein was not measured in the Zaghlool et al sample. Our results extend the work of Zaghlool et al further with the identification of several additional sites that are pleiotropic, such as the associations for SLC7A11 and PARP9 that were previously unmeasured. We have provided a more detailed view of the pleiotropy present in our study in the revised Figure 3 and Supplementary Tables 9-10. We have also added further clarification on the number of proteins that were previously uncharacterised by Zaghlool et al that we have analysed for the first time in our results section as follows:

Page 6, line 137: *'When accounting for 26 pQTMs that were previously reported by Zaghlool et al and 10 pQTMs that were previously reported by Hillary et al ^{14,19}, 2,892 of the 2,928 fully-adjusted pQTMs were novel. Of these 2,892 novel pQTMs, 1,109 involved the levels of 41 proteins that were measured by Zaghlool et al (973 pQTMs for PAPP A and 136 additional pQTMs for the levels of 40 proteins), whereas 1,783 pQTMs involved the levels of proteins that were previously unmeasured (1,116 pQTMs for PRG3 and 667 further pQTMs for 148 proteins).'*

Comment ten

10. What prompts the switch of multiple adjustment methods to FDR for the protein PheWAS rather than stick to one?

Response to comment ten

In our first draft of the manuscript we used conventions from previous MWAS and PheWAS studies. For example, Lehallier et al used FDR correction over their PheWAS results between SOMAmer measurements and age and sex. Zaghlool et al used the correction threshold of 0.05 / total proteins / total CpGs for their MWAS.

However, we have revisited the rationale for this. Since there is high correlation structure between the 4,235 protein measurements, we feel a Bonferroni threshold is somewhat harsh in this instance, given there are likely fewer independent components to the protein data. For this reason, we conducted a PCA analysis on the 4,235 SOMAmers in 1,065 individuals using `prcomp()` in R. This suggested that 143 independent components were able to explain >80% of the cumulative variance in protein levels (Supplementary Fig. 1 and Supplementary Table 3).

For this reason, we have chosen the following adjustment: MWAS: $0.05 / 143$ components * 772,619 CpGs = $P < 4.5 \times 10^{-10}$. However, we also consider the more stringent threshold previously used ($P < 0.05 / 4,235 * 772,619 = 1.5 \times 10^{-11}$) by demarcating the difference in associations between the previous and revised threshold in Supplementary Tables 4-6, which detail the MWAS results. While the updated MWAS threshold is slightly less stringent than that used previously, whether a pQTM falls into the more stringent threshold can therefore be easily referenced.

We have chosen the following threshold for the PheWAS: $0.05 / 143$ components = $P < 3.5 \times 10^{-4}$. When comparing adjustment in the PheWAS to our previous FDR-based method, we generally see that this is a more stringent approach, with the number of associations falling from 644 to 405. This is also evidenced by our age/sex PheWAS, in which the revised threshold reduced associations from 800 to 587 for age and 805 to 545 for sex, as compared to our original FDR $P < 0.05$ adjustment strategy (Supplementary Table 12). We therefore feel that this is appropriate. The high rate of replications between the age (97%) and sex (98%) associations in one or more of the three studies we use for comparisons (Feringstad et al 2021, Sun et al 2018 and Lehallier et al 2019) also supports this – see Supplementary Table 12 for this information, which is described as follows:

Page 7, line 154: 'When comparable associations from three studies (with $N > 1000$) were tested^{20,27,28}, 97% of age and 98% of sex associations replicated in one or more of studies (Supplementary Table 12).'

We have updated the manuscript as follows:

Page 5, line 95: '143 principal components explained 80% of the cumulative variance in the 4,235 measurements (Supplementary Fig. 1 and Supplementary Table 3). A threshold for multiple testing based on these components was applied across all MWAS ($P < 0.05/(143 \times 772,619) = 4.5 \times 10^{-10}$).'

Page 7, line 150: 'A threshold for multiple testing adjustment was calculated based on 143 independent components that explained >80% of the 4,235 SOMAmer levels (Supplementary Table 3 and Supplementary Fig. 1). This equated to $P < 0.05/(143) = 3.5 \times 10^{-4}$.'

Page 20, line 469: 'The Prcomp package⁷⁸ was used to generate principal components for the 4,235 SOMAmer measurements ($N=1,065$). 143 components explained >80% of the cumulative variance in protein levels (Supplementary Fig. 1 and Supplementary Table 3); these components were used to derive the PheWAS multiple testing adjustment threshold of $P < 0.05 / 143 = 3.5 \times 10^{-4}$. This method was chosen due to the presence of high intercorrelations within the protein data.'

Page 21, line 496: 'A threshold for multiple testing correction ($P < 4.5 \times 10^{-10}$) was based on 143 independent protein components with cumulative variance >80% (Supplementary Fig. 1 and Supplementary Table 3) ($P < 0.05/(143 \times 772,619)$ CpGs). A more conservative threshold based on total number of SOMAmers was also considered ($P < 0.05/(4,235 \times 772,619) = 1.5 \times 10^{-11}$) and is detailed in Supplementary Tables 4-6.'

Comment eleven

11. I believe other studies including Sun et al, Menni et al, Ngo et al also looked at association with age, gender, + others such as BMI/eGFR in addition to Lehallier et al.

Response to comment eleven

Thank you for drawing our attention to these studies. We collated the studies that examine age/sex associations and decided to include only those with N individuals greater than 1000 individuals that had looked at associations directly in a comparable format. A summary of inclusions/exclusions is provided below.

Chosen studies to include

- Sun et al, 2018 – (N 3,301 individuals)
- Ferkingstad et al, 2021 – (N 35,559 individuals)
- Lehallier et al, 2019 – (N 4,263 individuals)

Remaining studies that had N < 1000 individuals or looked at age/sex associations with DNAm mediation and were therefore not considered:

- Tanaka et al – (N 240 individuals)
<https://onlinelibrary.wiley.com/doi/full/10.1111/ace.12799>
- Tanaka et al – (N 997 individuals)
<https://elifesciences.org/articles/61073> (Age associated proteins mediation by DNA methylation)
- Ngo et al – (N < 100 individuals)
<https://www.ahajournals.org/doi/full/10.1161/CIRCULATIONAHA.116.021803>
- Menni et al – (N 206 individuals)
<https://academic.oup.com/biomedgerontology/article/70/7/809/707747?login=true#supplementary-data>

This replication assessment can be found in the updated Supplementary Table 12. Our manuscript also now reads as follows:

Page 7, line 152: *'The levels of 587 plasma proteins were associated with age and 545 were associated with sex, with 222 proteins common to both phenotypes (Supplementary Table 12). When comparable associations from three studies (with N > 1000) were tested^{20,27,28}, 97% of age and 98% of sex associations replicated in one or more of studies (Supplementary Table 12).'*

Comment twelve

12. The proteomic associations with other phenotypes: how much is known and how many are new? Maybe a forest plot with effect sizes for all/novel proteins rather than

an arbitrary selection of scatterplots would be more informative for the space in the figure?

Response to comment twelve

We have made the number of novel proteins for each trait clearer throughout the manuscript. Our replication assessment reads as follows:

Page 8, line 191: *'Six of the 14 APOE e4 status associations replicated previous SOMAmer protein findings (N SOMAmers= 4,785 and N participants=227)¹⁰, and eight novel relationships involved NEFL, ING4, PAF, MENT, TMCC3, CRP, FAM20A and PEF1. Several of the markers for cognitive function were identified in previous work relating Olink proteins to cognitive function (such as CPM)²⁹ and work that characterised SOMAmer signatures of cognitive decline and incident Alzheimer's disease (such as SVEP1)⁸. No studies have performed SOMAmer-based, whole proteome PheWAS studies of the brain imaging and cognitive score traits we have profiled in a healthy ageing population that were not enriched for neurodegenerative diseases. However, replication of associations from several studies^{9,29,30} was found for a small subset of associations (Supplementary Table 19).'*

We have also updated the revised Figure 4 such that it now details the number of total associations. As the number of novel associations would be too large to detail on a forest plot with ease, we have plotted all associations for APOE in Supplementary Figure 7. We also then include a replication summary in Supplementary Table 19 for the cognitive score and brain imaging replication assessment. We also provide the subset of 405 associations that were significant in the PheWAS as Supplementary Table 17. This means that future readers can easily search the summary statistics for the 405 associations presented by protein gene name in Supplementary Table 16.

Figure 4b shows a subset of protein associations that involved proteins that were markers for either a cognitive trait and APOE status (3 proteins), or a marker for a cognitive trait and brain imaging trait (22 proteins). The updated text reads as follows:

Page 8, line 181: *'Markers such as ASB9 and NCAN were found to be consistently identified across multiple brain imaging traits as markers of poorer and better brain health, respectively (Supplementary Table 16). While many of the associations for brain imaging measures identified proteins that were distinct from those found for cognitive scores and APOE e4 status, 22 protein markers were associated with both a cognitive score and a brain imaging trait (Fig. 4b and Supplementary Table 18). Of these 22 proteins, there were 10 principal components that had a cumulative variance of >80% and five components had eigenvalues > 1 (Supplementary Fig. 11). Three APOE e4 status markers (ING4, APOB and CRP) were also associated with cognitive scores (Fig. 4b).'*

Comment thirteen

13. "Many of the 644 protein marker associations were independent and did not cross neurological modalities." How is this determined?

Response to comment thirteen

Apologies that this was not clear in our original text. Simply put, we meant that many of the proteins markers we identified were not associated with multiple types of brain health outcome (i.e. associations for cognitive scores did not largely crossover with the associations for brain imaging, and only 3 APOE-associated proteins were associated with cognitive scores). We appreciate that the 'modalities' term is unclear and have updated our manuscript as follows:

Page 8, line 181: *'Markers such as ASB9 and NCAN were found to be consistently identified across multiple brain imaging traits as markers of poorer and better brain health, respectively (Supplementary Table 16). While many of the associations for brain imaging measures identified proteins that were distinct from those found for cognitive scores and APOE e4 status, 22 protein markers were associated with both a cognitive score and a brain imaging trait (Fig. 4b and Supplementary Table 18). Of these 22 proteins, there were 10 principal components that had a cumulative variance of >80% and five components had eigenvalues > 1 (Supplementary Fig. 11). Three APOE e4 status*

markers (*ING4*, *APOB* and *CRP*) were also associated with cognitive scores (Fig. 4b).'

Comment fourteen

14. . Of the 25 common proteins, there were six independent signals, as determined by components with eigenvalues > 1 in principal components analyses (Supplementary Fig.1). How is this justified? Why use eigenvalue of 1 – doesn't cumulative proportion essentially say the same thing? The first 6 PCs explain less than 70% of the variance. May be an alternative way to cluster may be needed here

Response to comment fourteen

We have updated our manuscript to clarify the number of clusters required to reach 80% of the cumulative variance explained, in addition to the components that had eigenvalues > 1 (a metric that is commonly applied to identify PCs of interest). Our results section reads as follows:

Page 8, line 186: *'Of these 22 proteins, there were 10 principal components that had a cumulative variance of $> 80\%$ and five components had eigenvalues > 1 (Supplementary Figure 11).'*

Comment fifteen

15. Are associated genes/proteins systematically enriched for any pathways from your main results?

Response to comment fifteen

We have added FUMA enrichment analyses and STRING protein interaction networks for the genes corresponding to the 191 unique proteins implicated in our PheWAS of brain health characteristics. These are provided as Supplementary Figures 8-10. We have also grouped the association summary in Supplementary Table 16 to include associations that reflected either poorer or more favourable brain health outcomes.

Our results section has been updated as follows:

Page 7, line 167: *'Stratifying the 405 associations by direction of effect revealed that the majority (89%) of associations indicated that higher levels of the proteins were associated with less favourable brain health (Supplementary Table 16). Eighty-seven of the 405 associations involved protein levels that were associated with more favourable brain health; this signature included the levels of SLITRK1, NCAN and COL11A2. Higher levels of ASB9, RBL2, HEXB and SMPD1 associated with poorer brain health. Protein interaction network analyses for the genes corresponding to the 191 protein markers (Supplementary Fig. 8) indicated that many of the proteins in these signatures clustered together, implying shared underlying functions. An inflammatory cluster including CRP, ITIH4, C3, C5, COL11A2 and SIGLEC2 was present and higher levels of these markers were associated with poorer brain health outcomes. Gene set enrichment analyses on the 191 genes corresponding to the protein markers (Supplementary Fig. 9) supported the link between many of the proteins associated with poorer brain health and the innate immune system, while also implicating extracellular matrix, lysosomal, metabolic and additional inflammatory pathways. Tissue expression profiles of the 191 genes (Supplementary Fig. 10) indicated that many of the markers were expressed non-neurological tissues; however, some proteins were expressed in nervous tissues. Markers such as ASB9 and NCAN were found to be consistently identified across multiple brain imaging traits as markers of poorer and better brain health, respectively (Supplementary Table 16).'*

Our discussion has been updated as follows:

Page 11, line 261: *'Many of the 191 proteins identified in the protein PheWAS were part of inflammatory clusters with shared functions in acute phase response, complement cascade activity, innate immune activity and cytokine pathways. Tissue expression analyses suggested that a large proportion of the 191 protein markers were not expressed in the brain; this supports work suggesting that sustained peripheral inflammation influences general brain health^{31,32} and accelerates cognitive decline^{8,33-35}. However, a subset of proteins were expressed in the central nervous system. Given that leakage at the blood-brain-barrier interface has been hallmarked as a part of healthy brain ageing^{36,37}, there is a possibility that brain-derived proteins may enter the bloodstream as biomarkers. SLIT and NTRK Like Family Member 1 (SLITRK1), Neurocan (NCAN) and IgLON family member 5 (IGLON5) were examples of proteins expressed in brain for which higher levels associated with either larger grey matter volume, larger whole*

*brain volume, or higher general fractional anisotropy. SLITRK1 localises at excitatory synapses and regulates synapse formation in hippocampal neurons*³⁸. *Neurocan (NCAN) is a component of neuronal extracellular matrix and is linked to neurite growth*³⁹. *IGLON5 has been implicated in maintenance of blood-brain-barrier integrity and an anti-IGLON5 antibody disease involves the deterioration of cognitive health*⁴⁰. *Taken together, the protein markers identified in the PheWAS may, therefore, reflect pathways that could be targeted to improve brain health.'*

We also include FUMA tissue expression and gene set enrichment analysis for the subset of 33 genes corresponding to either CpGs or proteins involved in neurological pQTM. These can be found in Supplementary Figures 13-14.

The following updates have been included:

Page 10, line 228: *'Tissue expression profiles for the 33 genes that were linked to either CpGs or proteins in the 35 neurological pQTMs are summarised in Supplementary Fig. 13. Gene set enrichment for these 33 genes identified enrichment for immune effector pathways in a subset of 11 genes, whereas a cluster of four genes (SMPD1, HEXB, AMY2A and AMY2B) were enriched for amylase and hydrolase activity (Supplementary Fig. 14).'*

Page 13, line 296: *'Many of the genes corresponding to CpGs and proteins involved in the 35 pQTMs were enriched for immune effector processes and were not expressed in brain. However, some markers did show evidence for brain-specific expression, such as acid sphingomyelinase (SMPD1) and Hexosaminidase Subunit Beta (HEXB).'*

Our methods section has been updated as follows, covering all FUMA and STRING analyses included:

Page 22, line 507: *'Functional mapping and annotation (FUMA)⁸² gene set enrichment analyses were conducted for genes corresponding to protein markers that were identified through the PheWAS study, in addition to genes linked to either CpGs or proteins in the neurological pQTM subset. Protein-coding genes were selected as the background set and ensemble v92 was used*

with a false discovery rate (FDR) adjusted $P < 0.05$ threshold for gene set testing. For the genes corresponding to protein markers in the PheWAS a minimum overlapping number of genes was set to 3, whereas this was set to 2 for the genes involved in neurological pQTM for the purposes of visualisation. The STRING⁸³ database was queried to build a protein interaction network based on all proteins that had associations in the PheWAS.'

REVIEWERS' COMMENTS

Reviewer #1 (Remarks to the Author):

The authors did a good job addressing my previous concerns. The manuscript has been significantly strengthened for publication in Nature Communications.

Reviewer #2 (Remarks to the Author):

The manuscript by Gadd et al. entitled "Integrated methylome and phenome study of the circulating proteome reveals markers pertinent to brain health" integrates DNA methylation and phenotypes uncovering multiple relationships and newer DNA methylation signatures in relation to brain health. The answers filled all the gaps found initially, and the manuscript is greatly improved. The authors have made a thorough revision of the manuscript and addressed the questions and concerns raised by the different reviewers and I have no additional comments.

Reviewer #4 (Remarks to the Author):

Thank you to the authors for addressing my comments (as much as the extent of availability of data allows) - I have no major outstanding concerns.

One minor comment:

"in addition to the components that had eigenvalues >1 (a metric that is commonly applied to identify PCs of interest)" - can you refer to the source/references that suggest this usage?

Can the necessary changes also be updated in the abstract to reflect the updated results please?

REVIEWER COMMENTS

General response to reviewers

We thank the reviewers for their positive feedback regarding the revisions. The work done to address their comments has, we believe, improved the rigour and quality of the work. As reviewers #1 and #2 did not have any further comments to address, we present a final point-by-point style response to the query raised by reviewer #4 below. Once again thank you to all reviewers for their time and expertise in reviewing this work.

Response to Reviewer #4

Thank you to the authors for addressing my comments (as much as the extent of availability of data allows) - I have no major outstanding concerns.

One minor comment:

"in addition to the components that had eigenvalues > 1 (a metric that is commonly applied to identify PCs of interest)" - can you refer to the source/references that suggest this usage?

Can the necessary changes also be updated in the abstract to reflect the updated results please?

Response to comment two part one

Thank you for raising this point regarding the use of thresholds for principal components analyses. While there are many possible sources to reference for the application of such thresholds, one appropriate source comes from work done by the statistician Ian Jolliffe.

In this publication titled 'Principal Component Analysis, Second Edition', on page 112 Jolliffe outlines that the dimensionality reduction thresholds we utilise in our paper (i.e. cumulative variance of 80% and eigenvalues > 1 as per Kaiser's rule) while being 'very much ad hoc rules-of-thumb' are 'intuitively plausible and that they work in practice'.

The full text can be accessed here:

[http://cda.psych.uiuc.edu/statistical_learning_course/Jolliffe%20I.%20Principal%20Component%20Analysis%20\(2ed.,%20Springer,%202002\)\(518s\)_MVsa .pdf](http://cda.psych.uiuc.edu/statistical_learning_course/Jolliffe%20I.%20Principal%20Component%20Analysis%20(2ed.,%20Springer,%202002)(518s)_MVsa.pdf).

Kaiser's rule is described on page 114 as follows: 'The rule, in its simplest form, is sometimes called Kaiser's rule (Kaiser, 1960) and retains only those PCs whose variances l_k exceed 1.'

The cumulative variance approach is described on page 112 as follows: 'Perhaps the most obvious criterion for choosing m , which has already been informally adopted in some of the examples of Chapters 4 and 5, is to select a (cumulative) percentage of total variation which one desires that the selected PCs contribute, say 80% or 90%. The

required number of PCs is then the smallest value of m for which this chosen percentage is exceeded.'

Professor Jolliffe's work on principal components analyses that we have referenced here has been cited in over 49,000 previous instances (see: <https://scholar.google.co.uk/citations?user=JdqoVEkAAAAJ&hl=en>).

We have therefore updated the text regarding principle analyses of the 22 protein marker levels that were associated with both a cognitive and imaging trait as follows:

Page 9, line 193: 'A principal components analysis of the 22 protein levels was conducted. The first five components had an eigenvalue > 1 and a cumulative variance of $>80\%$ was explained by the first 10 components. These are both commonly-used thresholds for deciding how many principal components to retain ²⁹.'

We have also updated the methods on the derivation of the 143 components across the full set of 4,235 protein measurements that we used to inform multiple testing correction as follows:

Page 22, line 498: 'The `Prcomp` function in the stats R package (Version 3.6.2) ⁷⁹ was used to generate principal components for the 4,235 SOMAmer measurements ($N=1,065$). 143 components explained $>80\%$ of the cumulative variance in protein levels (a commonly-used threshold for the retention of principal components ²⁹: Supplementary Fig. 1 and Supplementary Data 3). These 143 components were used to derive the PheWAS multiple testing adjustment threshold of $P < 0.05 / 143 = 3.5 \times 10^{-4}$. This method was chosen due to the presence of high intercorrelations within the protein data.'

These amends have been made in tracked changes in the manuscript file. They now detail the dimensionality reduction techniques used with further clarity and provide a reference to substantiate the thresholds used. While every statistical threshold is imperfect, these are commonly-used thresholds that serve to isolate components and we believe their use is warranted in these analyses.

Finally, regarding the request to add this update to the abstract, these methods details are not the primary analyses of this work. Therefore, given the limited text allowance of the abstract that is available, we hope it is acceptable to include full details in the results and methods section with referencing as we have done here. Additionally, it is not possible to reference in the abstract – the edits we have made to the main manuscript regarding this point are therefore better-substantiated.